# NODE-GAM:
## Neural Generalized Additive Model for Interpretable Deep Learning

[1,2,3]**Chun-Hao Chang**, [4]**Rich Caruana**, [1,2,3]**Anna Goldenberg**
[1]University of Toronto, [2]Vector Institute, [3]Hospital of Sickkids, [4]Microsoft Research
`kingsley@cs.toronto.edu, rcaruana@microsoft.com, anna.goldenberg@utoronto.ca`

### Abstract

Deployment of machine learning models in real high-risk settings (e.g. healthcare) often depends not only on the model's accuracy but also on its fairness, robustness, and interpretability. Generalized Additive Models (GAMs) are a class of interpretable models with a long history of use in these high-risk domains, but they lack desirable features of deep learning such as differentiability and scalability. In this work, we propose a neural GAM (NODE-GAM) and neural GA$^2$M (NODE-GA$^2$M) that scale well and perform better than other GAMs on large datasets, while remaining interpretable compared to other ensemble and deep learning models. We demonstrate that our models find interesting patterns in the data. Lastly, we show that we improve model accuracy via self-supervised pre-training, an improvement that is not possible for non-differentiable GAMs.

## 1 Introduction

As machine learning models become increasingly adopted in everyday life, we begin to require models to not just be accurate, but also satisfy other constraints such as fairness, bias discovery, and robustness under distribution shifts for high-stakes decisions (e.g., in healthcare, finance and criminal justice). These needs call for an easier ability to inspect and understand a model's predictions.

Generalized Additive Models (GAMs) (Hastie & Tibshirani, 1990) have a long history of being used to detect and understand tabular data patterns in a variety of fields including medicine (Hastie & Tibshirani, 1995; Izadi, 2020), business (Sapra, 2013) and ecology (Pedersen et al., 2019). Recently proposed tree-based GAMs and GA$^2$Ms models (Lou et al., 2013) further improve on original GAMs (Spline) having higher accuracy and better ability to discover data patterns (Caruana et al., 2015). These models are increasingly used to detect dataset bias (Chang et al., 2021) or audit black-box models (Tan et al., 2018a;b). As a powerful class of models, they still lack some desirable features of deep learning that made these models popular and effective, such as differentiability and scalability.

In this work, we propose a deep learning version of GAM and GA$^2$M that enjoy the benefits of both worlds. Our models are comparable to other deep learning approaches in performance on tabular data while remaining interpretable. Compared to other GAMs, our models can be optimized using GPUs and mini-batch training allowing for higher accuracy and more effective scaling on larger datasets. We also show that our models improve performance when labeled data is limited by self-supervised pretraining and finetuning, where other non-differentiable GAMs cannot be applied.

Several works have focused on building interpretable deep learning models that are effective for tabular data. TabNet (Arik & Pfister, 2020) achieves state-of-the-art performance on tabular data while also providing feature importance per example by its attention mechanism. Although attention seems to be correlated with input importance (Xu et al., 2015), in the worst case they might not correlate well (Wiegreffe & Pinter, 2019). Yoon et al. (2020) proposes to use self-supervised learning on tabular data and achieves state-of-the-art performance but does not address interpretability. NIT (Tsang et al., 2018) focuses on building a neural network that produces at most K-order interactions and thus include GAM and GA$^2$M. However, NIT requires a two-stage iterative training process that requires longer computations. And their performance is slightly lower to DNNs while ours are overall on par with it. They also do not perform purification that makes GA$^2$M graphs unique when showing them.

The most relevant approaches to our work are NODE (Popov et al., 2019) and NAM (Agarwal et al., 2020). Popov et al. (2019) developed NODE that mimics an ensemble of decision trees but permits differentiability and achieves state-of-the-art performance on tabular data. Unfortunately, NODE suffers from a lack of interpretability similarly to other ensemble and deep learning models. On the other hand, Neural Additive Model (NAM) whose deep learning architecture is a GAM, similar to our proposal, thus assuring interpretability. However, NAM can not model the pairwise interactions and thus do not allow GA$^2$M. Also, because NAM builds a small feedforward net per feature, in high-dimensional datasets NAM may require large memory and computation. Finally, NAM requires training of 10s to 100s of models and ensemble them which incurs large computations and memory, while ours only trains once; our model is also better than NAM without the ensemble (Supp. A).

To make our deep GAM scalable and effective, we modify NODE architecture (Popov et al., 2019) to be a GAM and GA$^2$M, since NODE achieves state-of-the-art performance on tabular data, and its tree-like nature allows GAM to learn quick, non-linear jumps that better match patterns seen in real data (Chang et al., 2021). We thus call our models NODE-GAM and NODE-GA$^2$M respectively.

One of our key contributions is that we design several novel gating mechanisms that gradually reduce higher-order feature interactions learned in the representation. This also enables our NODE-GAM and NODE-GA$^2$M to automatically perform feature selection via back-propagation for both marginal and pairwise features. This is a substantial improvement on tree-based GA$^2$M that requires an additional algorithm to select which set of pairwise feature interactions to learn (Lou et al., 2013).

Overall, our contributions can be summarized as follows:

- Novel architectures for neural GAM and GA$^2$M thus creating interpretable deep learning models.
- Compared to state-of-the-art GAM methods, our NODE-GAM and NODE-GA$^2$M achieve similar performance on medium-sized datasets while outperforming other GAMs on larger datasets.
- We demonstrate that NODE-GAM and NODE-GA$^2$M discover interesting data patterns.
- Lastly, we show that NODE-GAM benefits from self-supervised learning that improves performance when labeled data is limited, and performs better than other GAMs.

We foresee our novel deep learning formulation of the GAMs to be very useful in high-risk domains, such as healthcare, where GAMs have already proved to be useful but stopped short from being applied to new large data collections due to scalability or accuracy issues, as well as settings where access to labeled data is limited. Our novel approach also benefits the deep learning community by adding high accuracy interpretable models to the deep learning repertoire.

## 2 BACKGROUND

**GAM and GA$^2$M:** GAMs and GA$^2$Ms are interpretable by design because of their functional forms. Given an input $x \in \mathbb{R}^D$, a label $y$, a link function $g$ (e.g. $g$ is $\log \frac{p}{1-p}$ in binary classification), main effects $f_j$ for each feature $j$, and feature interactions $f_{jj'}$, GAM and GA$^2$M are expressed as:

$$\textbf{GAM}: \quad g(y) = f_0 + \sum_{j=1}^{D} f_j(x_j), \quad \textbf{GA}^2\textbf{M}: \quad g(y) = f_0 + \sum_{j=1}^{D} f_j(x_j) + \sum_{j=1}^{D} \sum_{j'>j} f_{jj'}(x_j, x_{j'}).$$

Unlike full complexity models (e.g. DNNs) that have $y = f(x_1, ..., x_j)$, GAMs and GA$^2$M are interpretable because the impact of each feature $f_j$ and each feature interaction $f_{jj'}$ can be visualized as a graph (i.e. for $f_j$, x-axis shows $x_j$ and y-axis shows $f_j(x_j)$). Humans can easily simulate how they work by reading $f_j$s and $f_{jj'}$ off different features from the graph and adding them together.

**GAM baselines:** We compare with Explainable Boosting Machine (EBM) (Nori et al., 2019) that implements tree-based GAM and GA$^2$M. We also compare with splines proposed in the 80s (Hastie & Tibshirani, 1990) using Cubic splines in pygam package (Servén & Brummitt, 2018).

**Neural Oblivious Decision Trees (NODEs):** We describe NODEs for completeness and refer the readers to Popov et al. (2019) for more details. NODE consists of $L$ layers where each layer has $I$ differentiable oblivious decision trees (ODT) of equal depth $C$. Below we describe a single ODT.

**Differentiable Oblivious Decision Trees:** An ODT works like a traditional decision tree except for all nodes in the same depth share the same input features and thresholds, which allows parallel computation and makes it suitable for deep learning. Specifically, an ODT of depth $C$ compares $C$ chosen input feature to $C$ thresholds, and returns one of the $2^C$ possible responses. Mathmatically, for feature functions $F^c$ which choose what features to split, splitting thresholds $b^c$, and a response vector $\boldsymbol{R} \in \mathbb{R}^{2^C}$, the tree output $h(\boldsymbol{x})$ is defined as:

$$h(\boldsymbol{x}) = \boldsymbol{R} \cdot \left( \begin{bmatrix} \mathbb{I}(F^1(\boldsymbol{x}) \leq b^1) \\ \mathbb{I}(F^1(\boldsymbol{x}) > b^1) \end{bmatrix} \otimes \begin{bmatrix} \mathbb{I}(F^2(\boldsymbol{x}) \leq b^2) \\ \mathbb{I}(F^2(\boldsymbol{x}) > b^2) \end{bmatrix} \otimes \cdots \otimes \begin{bmatrix} \mathbb{I}(F^C(\boldsymbol{x}) \leq b^C) \\ \mathbb{I}(F^C(\boldsymbol{x}) > b^C) \end{bmatrix} \right) \tag{1}$$

Here $\mathbb{I}$ is the indicator function, $\otimes$ is the outer product, and $\cdot$ is the inner product.

Both feature functions $F^c$ and $\mathbb{I}$ prevent differentiability. To make them differentiable, Popov et al. (2019) replace $F^c(\boldsymbol{x})$ as a weighted sum of features:

$$F^c(\boldsymbol{x}) = \sum_{j=1}^{D} x_j \text{entmax}_\alpha(F^c)_j = \boldsymbol{x} \cdot \text{entmax}_\alpha(\boldsymbol{F}^c). \tag{2}$$

Here $\boldsymbol{F}^c \in \mathbb{R}^D$ are the logits for which features to choose, and $\text{entmax}_\alpha$ (Peters et al., 2019) is the entmax transformation which works like a sparse version of softmax such that the sum of the output equals to 1. They also replace the $\mathbb{I}$ with entmoid which works like a sparse sigmoid that has output values between 0 and 1. Since all operations are differentiable (entmax, entmoid, outer and inner products), the ODT is differentiable.

**Stacking trees into deep layers:** Popov et al. (2019) follow the design similar to DenseNet where all tree outputs $\boldsymbol{h}(\boldsymbol{x})$ from previous layers (each layer consists of total $I$ trees) become the inputs to the next layer. For input features $\boldsymbol{x}$, the inputs $\boldsymbol{x}^l$ to each layer $l$ becomes:

$$\boldsymbol{x}^1 = \boldsymbol{x}, \quad \boldsymbol{x}^l = [\boldsymbol{x}, \boldsymbol{h}^1(\boldsymbol{x}^1), ..., \boldsymbol{h}^{(l-1)}(\boldsymbol{x}^{(l-1)})] \text{ for } l > 1. \tag{3}$$

And the final output of the model $\hat{y}(\boldsymbol{x})$ is the average of all tree outputs $\boldsymbol{h}_1, ..., \boldsymbol{h}_L$ of all $L$ layers:

$$\hat{y}(x) = \frac{1}{LI} \sum_{l=1}^{L} \sum_{i=1}^{I} h_{li}(\boldsymbol{x}^l) \tag{4}$$

## 3 OUR MODEL DESIGN

**GAM design:** See Supp. C for a complete pseudo code. To make NODE a GAM, we make three key changes to avoid any feature interactions in the architecture (Fig. 1). First, instead of letting $F^c(\boldsymbol{x})$ be a weighted sum of features (Eq. 2), we make it only pick 1 feature. We introduce a temperature annealing parameter $T$ that linearly decreases from 1 to 0 for the first $S$ learning steps to make $\text{entmax}_\alpha(\boldsymbol{F}^c/T)$ gradually become one-hot:

$$F^c(\boldsymbol{x}) = \boldsymbol{x} \cdot \text{entmax}_\alpha(\boldsymbol{F}^c/T), \quad T \xrightarrow{S \text{ steps}} 0. \tag{5}$$

Second, within each tree, we make the logits $\boldsymbol{F}^c$ the same across depth $C$ i.e. $\boldsymbol{F}^1 = \cdots = \boldsymbol{F}^C = \boldsymbol{F}$ to avoid any feature interaction within a tree. Third, we avoid the DenseNet connection between two trees that focus on different features $j, j'$, since they create feature interactions between features $j$ and $j'$ if two trees connect. Thus we introduce a gate that only allows connections between trees that take the same features. Let $\boldsymbol{G}_i = \text{entmax}_\alpha(\boldsymbol{F}_i/T)$ of the tree $i$. For tree $i$ in layer $l$ and another tree $\hat{i}$ in layer $\hat{l}$ for $\hat{l} < l$, the gating weight $g_{\hat{l}\hat{i}i}$ and the feature function $F_{li}$ for tree $i$ become:

$$g_{\hat{l}\hat{i}i} = \boldsymbol{G}_{\hat{i}} \cdot \boldsymbol{G}_i, \quad F_{li}(\boldsymbol{x}) = \boldsymbol{x} \cdot \boldsymbol{G}_i + \frac{1}{\sum_{\hat{l}=1}^{l-1} \sum_{\hat{i}=1}^{I} g_{\hat{l}\hat{i}i}} \sum_{\hat{l}=1}^{l-1} \sum_{\hat{i}=1}^{I} h_{\hat{l}\hat{i}}(\boldsymbol{x}) g_{\hat{l}\hat{i}i}. \tag{6}$$

Since $\boldsymbol{G}$ becomes gradually one-hot by Eq. 5, after $S$ steps $g_{\hat{i}i}$ would only become 1 when $\boldsymbol{G}_{\hat{i}} = \boldsymbol{G}_i$ and 0 otherwise. This enforces no feature interaction between tree connections.

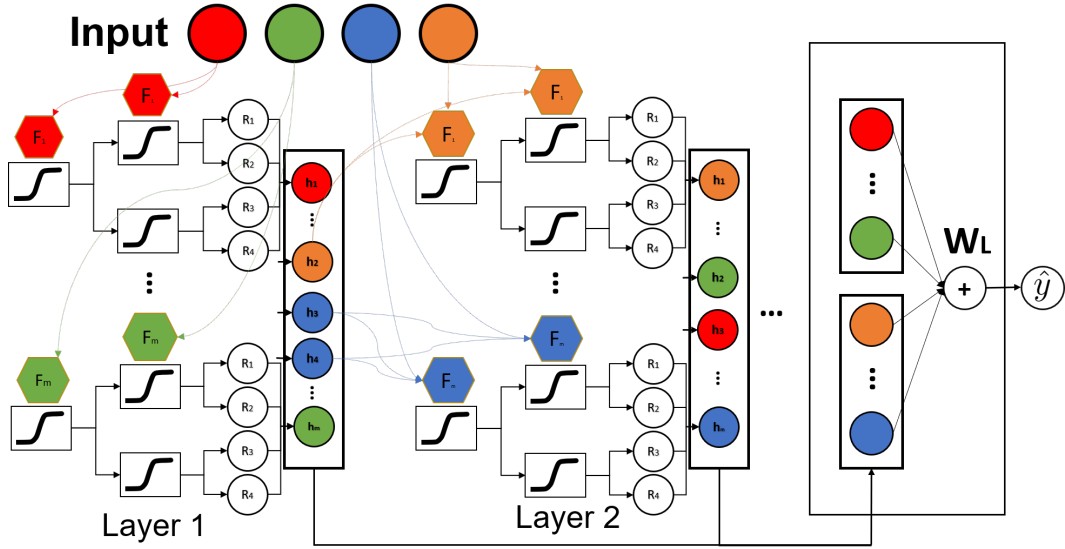

Figure 1: The NODE-GAM architecture. Here we show $4$ features with $4$ different colors. Each layer consists of $I$ differentiable oblivious decision trees that outputs $h_1...h_I$, where each $h_i$ only depends on $1$ feature. We only connect trees between layers if two trees depend on the same features. And we concatenate all outputs from all layers as inputs to the last linear layer $\boldsymbol{W}_L$ to produce outputs.

**Attention-based GAMs (AB-GAMs):** To make the above GAM more expressive, we add an attention weight $a_{l\hat{i}i}$ in the feature function $F_{li}(x)$ to decide which previous tree to focus on:

$$F_{li}(\boldsymbol{x}) = \sum_{j=1}^{D} x_j G_{ij} + \sum_{\hat{l}=1}^{l-1} \sum_{\hat{i}=1}^{I} h_{\hat{l}\hat{i}}(x) g_{l\hat{i}i} a_{l\hat{i}i} \quad \text{where} \quad \sum_{\hat{l}=1}^{l-1} \sum_{\hat{i}=1}^{I} g_{l\hat{i}i} a_{l\hat{i}i} = 1. \tag{7}$$

To achieve this, we introduce attention logits $\boldsymbol{A_{li}}$ for each tree $i$ that after entmax it produces $a_{l\hat{i}i}$:

$$a_{l\hat{i}i} = g_{l\hat{i}i}\text{entmax}_\alpha(\log(\boldsymbol{g}_i) + \boldsymbol{A}_{li})_{\hat{i}}. \tag{8}$$

This forces the attention of a tree $i$ that $\sum_{\hat{i}} a_{l\hat{i}i} = 1$ for all $\hat{i}$ that $g_{l\hat{i}i} = 1$ and $a_{l\hat{i}i} = 0$ when $g_{l\hat{i}i} = 0$.

The attention logits $\boldsymbol{A}$ requires a large matrix size $[I, (l-1)I]$ for each layer $l > 1$ which explodes the memory. We instead make $A$ as the inner product of two smaller matrices such that $A = BC$ where B is of size $[I, E]$ and C is of size $[E, (l-1)I]$, where $E$ is a hyperparameter for the embedding dimension of the attention.

**Last Linear layer:** Lastly, instead of averaging the outputs of all trees as the output of the model (Eq. 4), we add the last linear layer to be a weighted sum of all outputs:

$$\hat{y}(x) = \sum_{l=1}^{L} \sum_{i=1}^{I} h_{li}(\boldsymbol{x_l}) w_{li}. \tag{9}$$

Note that in self-supervised learning, $w_{li}$ has multiple output heads to predict multiple tasks.

**Regularization:** We also include other changes that improves performance. First, we add Dropout (rate $p1$) on the outputs of trees $h_{li}(\boldsymbol{x_l})$, and Dropout (rate $p2$) on the final weights $w_{li}$. Also, to increase diversity of trees, each tree can only model on a random subset of features ($\eta$), an idea similar to Random Forest. We also add an $\ell_2$ penalization ($\lambda$) on $h_{li}(\boldsymbol{x_l})$. In binary classification task where labels $y$ are imbalanced between class 0 and 1, we set a constant as $\log \frac{p(y)}{1-p(y)}$ that is added to the final output of the model such that after sigmoid it becomes the $p(y)$ if the output of the model is 0. We find it's crucial for $\ell_2$ penalization to work since $\ell_2$ induces the model to output 0.

**NODE-GA$^2$Ms — extending NODE-GAMs to two-way interactions:** To allow two-way interactions, for each tree we introduce two logits $\boldsymbol{F}^1$ and $\boldsymbol{F}^2$ instead of just one, and let $\boldsymbol{F}^c = \boldsymbol{F}^{(c-1)\bmod 2+1}$ for $c > 2$; this allows at most 2 features to interact within each tree (Fig. 7). Besides temperature annealing (Eq. 5), we make the gating weights $g_{\hat{i}i} = 1$ only if the combination of $\boldsymbol{F}^1, \boldsymbol{F}^2$ is the same between tree $\hat{i}$ and $i$ (i.e. both trees $\hat{i}$ and $i$ focus on the same 2 features). We set $g_{\hat{i}i}$ as:

$$g_{\hat{i}i} = min((\boldsymbol{G}_i^1 \cdot \boldsymbol{G}_{\hat{i}}^1) \times (\boldsymbol{G}_i^2 \cdot \boldsymbol{G}_{\hat{i}}^2) + (\boldsymbol{G}_i^1 \cdot \boldsymbol{G}_{\hat{i}}^2) \times (\boldsymbol{G}_i^2 \cdot \boldsymbol{G}_{\hat{i}}^1), 1). \tag{10}$$

We cap the value at 1 to avoid uneven amplifications as $g_{\hat{i}i} = 2$ when $\boldsymbol{G}_i^1 = \boldsymbol{G}_i^2 = \boldsymbol{G}_{\hat{i}}^1 = \boldsymbol{G}_{\hat{i}}^2$.

**Data Preprocessing and Hyperparameters:** We follow Popov et al. (2019) to do target encoding for categorical features, and do quantile transform for all features to Gaussian distribution (we find Gaussian works better than Uniform). We use random search to search the architecture space for NODE, NODE-GAM and NODE-GA$^2$M. We use QHAdam (Ma & Yarats, 2018) and average the most recent 5 checkpoints (Izmailov et al., 2018). In addition, we adopt learning rate warmup (Goyal et al., 2017), and do early stopping and learning rate decay on the plateau. More details in Supp. G.

**Extracting shape graphs from GAMs:** We follow Chang et al. (2021) to implement a function that extracts main effects $f_j$ from any GAM model including NODE-GAM, Spline and EBM. The main idea is to take the difference between the model's outputs of two examples $(\boldsymbol{x}^1, \boldsymbol{x}^2)$ that have the same values except for feature $j$. Since the intercept and other main effects are canceled out when taking the difference, the difference $f(x^2) - f(x^1)$ is equal to $f_j(x_j^2) - f_j(x_j^1)$. If we query all the unique values of $x_j$, we get all values of $f_j$ relative to $f_j(x_j^1)$. Then we center the graph of $f_j$ by setting the average of $f_j(x_j)$ across the dataset as 0 and add the average to the intercept term $f_0$.

**Extracting shape graphs from GA$^2$Ms:** Designing a black box function to extract from any GA$^2$M is non-trivial, as each changed feature $x_j$ would change not just main effect term $f_j$ but also every interactions $\forall_{j'} f_{jj'}$ that involve feature $j$. Instead, since we know which features each tree takes, we can aggregate the output of trees into corresponding main $f_j$ and interaction terms $f_{jj'}$.

Note that GA$^2$M can have many representations that result in the same function. For example, for a prediction value $v$ associated with $x_2$, we can move $v$ to the main effect $f_2(x_2) = v$, or the interaction effect $f_{23}(x_2, \cdot) = v$ that involves $x_2$. To solve this ambiguity, we adopt "purification" (Lengerich et al., 2020) that pushes interaction effects into main effects if possible. See Supp. D for details.

## 4 RESULTS

We first show the accuracy of our models in Sec. 4.1. Then we show the interpretability of our models on Bikeshare and MIMIC2 datasets in Sec. 4.2. In Sec. 4.3, we show that NODE-GAM benefits from self-supervised pre-training and outperforms other GAMs when labels are limited. In Supp. A, we show our model outperforms NAM without ensembles. In Supp. B, we provide a strong default hyperparameter that still outperforms EBM without hyperparameter tuning.

### 4.1 ARE NODE-GAM AND NODE-GA$^2$M ACCURATE?

We compare our performance on 6 popular binary classification datasets (Churn, Support2, MIMIC2, MIMIC3, Income, and Credit) and 2 regression datasets (Wine and Bikeshare). These datasets are medium-sized with 6k-300k samples and 6-57 features (Table 6). We use 5-fold cross validation to derive the mean and standard deviation for each model. We use 80-20 splits for training and val set. To compare models across datasets, we calculate 2 summary metrics: (1) Rank: we rank the performance on each dataset, and then compute the average rank across all 9 datasets (the lower the rank the better). (2) Normalized Score (NS): for each dataset, we set the worst performance for that dataset as 0 and the best as 1, and scale all other scores linearly between 0 and 1.

In Table 1, we show the performance of all GAMs, GA$^2$Ms and full complexity models. First, we compare 4 GAMs (here NODE-GA$^2$M-main is the purified main effect from NODE-GA$^2$M). We find all 4 GAMs perform similarly and the best GAM in different datasets varies, with Spline as the best in Rank and NODE-GAM in NS. But the differences are often smaller than the standard deviation.

Table 1: The performance for 8 medium-sized datasets. The first 6 datasets are binary classification (ordered by samples) and shown the AUC (%). The last 2 are regression datasets and shown the Root Mean Squared Error (RMSE). We show the standard deviation of 5-fold cross validation results. We calculate average rank (Rank, lower the better) and average Normalized Score (NS, higher the better).

| | GAM | | | | GA$^2$M | | Full Complexity | | |
|---|---|---|---|---|---|---|---|---|---|
| | NODE GAM | NODE GA$^2$M Main | EBM | Spline | NODE GA$^2$M | EBM GA$^2$M | NODE | XGB | RF |
| Churn | 84.9$_{\pm 0.8}$ | 84.9$_{\pm 0.9}$ | 85.0$_{\pm 0.7}$ | **85.1**$_{\pm 0.9}$ | 85.0$_{\pm 0.8}$ | 85.0$_{\pm 0.7}$ | 84.3$_{\pm 0.6}$ | 84.7$_{\pm 0.9}$ | 82.9$_{\pm 0.8}$ |
| Support2 | 81.5$_{\pm 1.3}$ | 81.5$_{\pm 1.1}$ | 81.5$_{\pm 1.0}$ | 81.5$_{\pm 1.1}$ | **82.7**$_{\pm 0.7}$ | 82.6$_{\pm 1.1}$ | **82.7**$_{\pm 1.0}$ | 82.3$_{\pm 1.0}$ | 82.1$_{\pm 1.0}$ |
| Mimic2 | 83.2$_{\pm 1.1}$ | 83.4$_{\pm 1.3}$ | 83.5$_{\pm 1.1}$ | 82.5$_{\pm 1.1}$ | 84.6$_{\pm 1.1}$ | 84.8$_{\pm 1.2}$ | 84.3$_{\pm 1.1}$ | 84.4$_{\pm 1.2}$ | **85.4**$_{\pm 1.3}$ |
| Mimic3 | 81.4$_{\pm 0.5}$ | 81.0$_{\pm 0.6}$ | 80.9$_{\pm 0.4}$ | 81.2$_{\pm 0.4}$ | 82.2$_{\pm 0.7}$ | 82.1$_{\pm 0.4}$ | **82.8**$_{\pm 0.7}$ | 81.9$_{\pm 0.4}$ | 79.5$_{\pm 0.7}$ |
| Income | 92.7$_{\pm 0.3}$ | 91.8$_{\pm 0.5}$ | 92.7$_{\pm 0.3}$ | 91.8$_{\pm 0.3}$ | 92.3$_{\pm 0.3}$ | **92.8**$_{\pm 0.3}$ | 91.9$_{\pm 0.3}$ | 92.8$_{\pm 0.3}$ | 90.8$_{\pm 0.2}$ |
| Credit | 98.1$_{\pm 1.1}$ | 98.4$_{\pm 1.0}$ | 97.4$_{\pm 0.9}$ | 98.2$_{\pm 1.1}$ | **98.6**$_{\pm 1.0}$ | 98.2$_{\pm 0.6}$ | 98.1$_{\pm 0.9}$ | 97.8$_{\pm 0.9}$ | 94.6$_{\pm 1.8}$ |
| Wine | 0.71$_{\pm 0.03}$ | 0.70$_{\pm 0.02}$ | 0.70$_{\pm 0.02}$ | 0.72$_{\pm 0.02}$ | 0.67$_{\pm 0.02}$ | 0.66$_{\pm 0.01}$ | 0.64$_{\pm 0.01}$ | 0.75$_{\pm 0.03}$ | **0.61**$_{\pm 0.01}$ |
| Bikeshare | 100.7$_{\pm 1.6}$ | 100.7$_{\pm 1.4}$ | 100.0$_{\pm 1.4}$ | 99.8$_{\pm 1.4}$ | 49.8$_{\pm 0.8}$ | 50.1$_{\pm 0.8}$ | **36.2**$_{\pm 1.9}$ | 49.2$_{\pm 0.9}$ | 42.2$_{\pm 0.7}$ |
| Rank | 5.8 | 6.2 | 5.9 | 5.3 | **3.2** | 3.5 | 4.5 | 3.9 | 6.6 |
| NS | 0.533 | 0.471 | 0.503 | 0.464 | 0.808 | **0.812** | 0.737 | 0.808 | 0.301 |

Table 2: The performance for 6 large datasets used in NODE paper. The first 3 datasets (Click, Epsilon and Higgs) are classification datasets and shown the Error Rate. The last 3 (Microsoft, Yahoo and Year) are shown in Mean Squared Error (MSE). We show the relative improvement (Rel Imp) of our model NODE-GAM to EBM and find it consistently outperforms EBM up to 7%.

| | GAM | | | | GA$^2$M | | | Full Complexity | | |
|---|---|---|---|---|---|---|---|---|---|---|
| | NODE GAM | EBM | Spline | Rel Imp | NODE GA$^2$M | EBM GA$^2$M | Rel Imp | NODE | XGB | RF |
| Click | 0.3342 $\pm$ 0.0001 | 0.3328 $\pm$ 0.0001 | 0.3369 $\pm$ 0.0002 | -0.4% | 0.3307 $\pm$ 0.0001 | 0.3297 $\pm$ 0.0001 | -0.2% | 0.3312 $\pm$ 0.0002 | 0.3334 $\pm$ 0.0002 | 0.3473 $\pm$ 0.0001 |
| Epsilon | 0.1040 $\pm$ 0.0003 | - | - | - | 0.1050 $\pm$ 0.0002 | - | - | 0.1034 $\pm$ 0.0003 | 0.1112 $\pm$ 0.0006 | 0.2398 $\pm$ 0.0008 |
| Higgs | 0.2970 $\pm$ 0.0001 | 0.3006 $\pm$ 0.0002 | - | 1.2% | 0.2566 $\pm$ 0.0003 | 0.2767 $\pm$ 0.0004 | 7.3% | 0.2101 $\pm$ 0.0005 | 0.2328 $\pm$ 0.0003 | 0.2406 $\pm$ 0.0001 |
| Microsoft | 0.5821 $\pm$ 0.0004 | 0.5890 $\pm$ 0.0006 | - | 1.2% | 0.5618 $\pm$ 0.0003 | 0.5780 $\pm$ 0.0001 | 2.8% | 0.5570 $\pm$ 0.0002 | 0.5544 $\pm$ 0.0001 | 0.5706 $\pm$ 0.0006 |
| Yahoo | 0.6101 $\pm$ 0.0006 | 0.6082 $\pm$ 0.0011 | - | -0.3% | 0.5807 $\pm$ 0.0004 | 0.6032 $\pm$ 0.0005 | 3.7% | 0.5692 $\pm$ 0.0002 | 0.5420 $\pm$ 0.0004 | 0.5598 $\pm$ 0.0003 |
| Year | 85.09 $\pm$ 0.01 | 85.81 $\pm$ 0.11 | - | 0.8% | 79.57 $\pm$ 0.12 | 83.16 $\pm$ 0.01 | 4.3% | 76.21 $\pm$ 0.12 | 78.53 $\pm$ 0.09 | 86.61 $\pm$ 0.06 |
| Average | - | - | - | 0.5% | - | - | 3.6% | - | - | - |

Next, both NODE-GA$^2$M and EBM-GA$^2$M perform similarly, with NODE-GA$^2$M better in Rank and EBM-GA$^2$M better in NS. Lastly, within all full complexity methods, XGB performs the best with not much difference from NODE and RF performs the worst. In summary, all GAMs perform similarly. NODE-GA$^2$M is similar to EBM-GA$^2$M, and slightly outperforms full-complexity models.

In Table 2, we test our methods on 6 large datasets (all have samples > 500K) used in the NODE paper, and we use the same train-test split to be comparable. Since these only provide 1 test split we report standard deviation across multiple random seeds. First, on a cluster with 32 CPU and 120GB memory, Spline goes out of memory on 5 out of 6 datasets and EBM also can not be run on dataset Epsilon with 2k features, showing their lack of ability to scale to large datasets. For 5 datasets that EBM can run, our NODE-GAMs runs slightly better than EBM. But when considering GA$^2$M, NODE-GA$^2$M outperforms EBM-GA$^2$M up to 7.3% in Higgs and average relative improvement of 3.6%. NODE outperforms all GAMs and GA$^2$Ms substantially on Higgs and Year, suggesting both datasets might have important higher-order feature interactions.

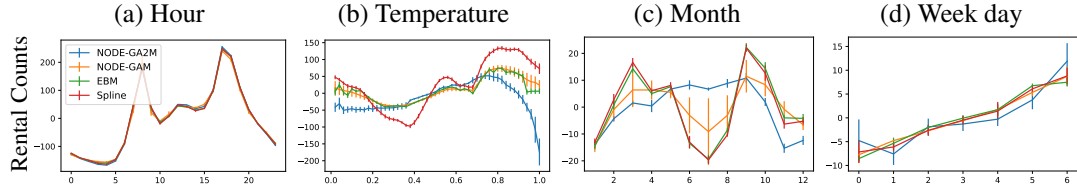

Figure 2: The shape plots of 4 (out of 11) features of 4 models (NODE-GA$^2$M, NODE-GAM, EBM, and Spline) trained on the Bikeshare dataset.

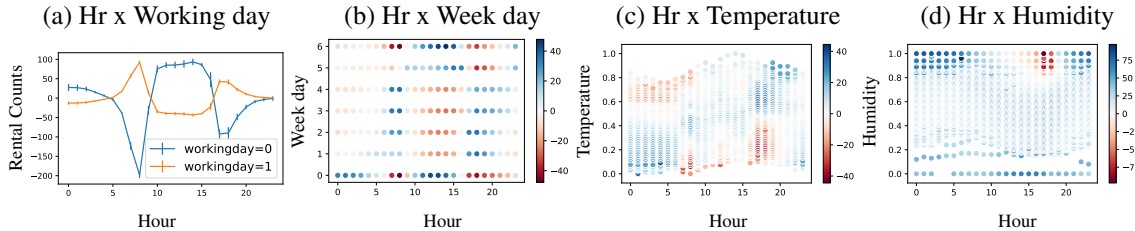

Figure 3: The shape plots of 4 interactions of NODE-GA$^2$M trained on the Bikeshare dataset.

## 4.2 SHAPE GRAPHS OF NODE-GAM AND NODE-GA$^2$M: BIKESHARE AND MIMIC2

In this section, we highlight our key findings, and show the rest of the plots in Supp. I.

**Bikeshare dataset:** Here we interpret the Bikeshare dataset. It contains the hourly count of rental bikes between the years 2011 and 2012 in Capital bikeshare system located in Washington, D.C. Note that all 4 GAMs trained on Bikeshare are equally accurate with $< 0.1\%$ error difference (Table 1).

In Fig. 2, we show the shape plots of 4 features: Hour, Temperature, Month, and Week day. First, Hour (Fig. 2a) is the strongest feature with two peaks around 9 AM and 5 PM, representing the time that people commute, and all 4 models agree. Then we show Temperature in Fig. 2b. Here temperature is normalized between 0 and 1, where 0 means -8°C and 1 means 39°C. When the weather is hot (Temp > 0.8, around 30°C), all models agree rental counts decrease which makes sense. Interestingly, when it's getting colder (Temp < 0.4, around 11°C) there is a steady rise shown by NODE-GAM, Spline and EBM but not NODE-GA$^2$M (blue). Since it's quite unlikely people rent more bikes when it's getting colder especially below 0°C, the pattern shown by GA$^2$M seems more plausible. Similarly, in feature Month (Fig. 2c), NODE-GA$^2$M shows a rise in summer (month $6 - 8$) while others indicate a strong decline of rental counts. Since we might expect more people to rent bikes during summer since it's warmer, NODE-GA$^2$M might be more plausible, although we might explain it due to summer vacation fewer students ride bikes to school. Lastly, for Weekday (Fig. 2d) all 4 models agree with each other that the lowest number of rentals happen at the start of the week (Sunday and Monday) and slowly increase with Saturday as the highest number.

In Fig. 3, we show the 4 feature interactions (out of 67) from our NODE-GA$^2$M. The strongest effect happens in Hr x Working day (Fig. 3(a)): this makes sense since in working day (orange), people usually rent bikes around 9AM and 5PM to commute. Otherwise, if the working day is 0 (blue), the number peaks from 10AM to 3PM which shows people going out more often in daytime. In Hr x Weekday (Fig. 3(b)), we can see more granularly that this commute effect happens strongly on Monday to Thursday, but on Friday people commute a bit later, around 10 AM, and return earlier, around 3 or 4 PM. In Hr x Temperature (Fig. 3(c)), it shows that in the morning rental count is high when it's cold, while in the afternoon the rental count is high when it's hot. We also find in Hr x Humidity (Fig. 3(d)) that when humidity is high from 3-6 PM, people ride bikes less. Overall these interpretable graphs enable us to know how the model predicts and find interesting patterns.

**MIMIC2:** MIMIC2 is the hospital ICU mortality prediction task (Johnson et al., 2016a). We extract 17 features within the first 24 hour measurements, and we use mean imputation for missingness.

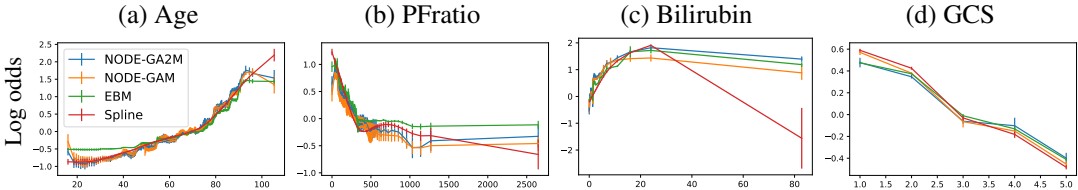

Figure 4: The shape plots of 4 GAMs trained on MIMIC-II dataset (4 of the 17 features are shown).

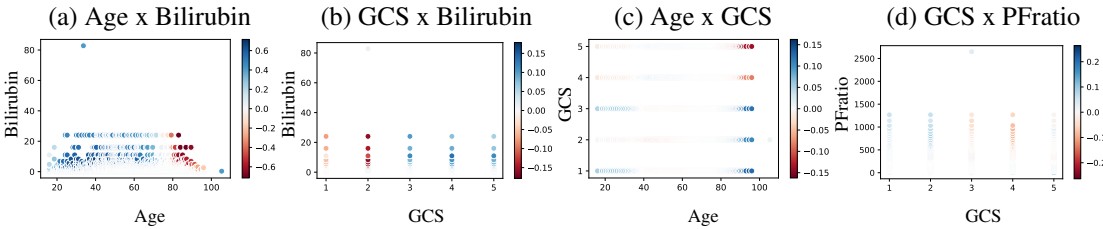

Figure 5: The shape plots of 4 interactions of NODE-GA$^2$M trained on the MIMIC2 dataset.

We show the shape plots (Fig. 4) of 4 features: Age, PFratio, Bilirubin and GCS. In feature Age (Fig.. 4(a)), we see the all 4 models agree the risk increases from age 20 to 90. Overall NODE-GAM/GA$^2$M are pretty similar to EBM in that they all have small jumps in a similar place at age 55, 80 and 85; spline (red) is as expected very smooth. Interestingly, we see NODE-GAM/GA$^2$M shows risk increases a bit when age < 20. We think the risk is higher in younger people because this is generally a healthier age in the population, so their presence in ICU indicates higher risk conditions.

In Fig. 4(b), we show PFratio: a measure of how well patients oxygenate the blood. Interestingly, NODE-GAM/GA$^2$M and EBM capture a sharp drop at 332. It turns out that PFratio is usually not measured for healthier patients, and the missing values have been imputed by the population mean 332, thus placing a group of low-risk patients right at the mean value of the feature. However, Spline (red) is unable to capture this and instead have a dip around 300-600. Another drop captured by NODE-GAM/GA$^2$M from $400 - 500$ matches clinical guidelines that $> 400$ is healthy.

Bilirubin shape plot is shown in Fig. 4(c). Bilirubin is a yellowish pigment made during the normal breakdown of red blood cells. High bilirubin indicates liver or bile duct problems. Indeed, we can see risk quickly goes up as Bilirubin is $> 2$, and all 4 models roughly agree with each other except for Spline which has much lower risk when Billirubin is 80, which is likely caused by Spline's smooth inductive bias and unlikely to be true. Lastly, in Fig. 4(d) we show Glasgow Coma Scale (GCS): a bedside measurement for how conscious the patient is with 1 in a coma and 5 as conscious. Indeed, we find the risk is higher for patients with GCS=1 than 5, and all 4 models agree.

In Fig. 5, we show the 4 of 154 feature interactions learned in the NODE-GA$^2$M. First, in Age x Bilirubin (Fig. 5(a)), when Billirubin is high (>2), we see an increase of risk (blue) in people with age $18 - 70$. Risk decreases (red) when age > 80. Combined with the shape plots of Age (Fig. 4(a)) and Bilirubin (Fig. 4(c)), we find this interaction works as a correction effect: if patients have Bilirubin > 2 (high risk) but are young (low risk), they should have a higher risk than what their main (univariate) effects suggest. On the other hand, if patients have age > 80 (high risk) and Bilirubin > 2 (high risk), they already get very high risk from main effects, and in fact the interaction effect is negative to correct for the already high main effects. It suggests that Billirubin=2 is an important threshold that should affect risk adjustments.

Also in GCS x Bilirubin plot (Fig. 5(b)), we find similar effects: if Bilirubin >2, the risk of GCS is correctly lower for GCS=1,2 and higher for 3-5. In Fig. 5(c) we find patients with GCS=1-3 (high risk) and age>80 (high risk), surprisingly, have even higher risk (blue) for these patients; it shows models think these patients are more in danger than their main effects suggest. Finally, in Fig. 5(d) we show interaction effect GCS x PFratio. We find PFratio also has a similar threshold effect: if PFratio > 400 (low risk), and GCS=1,2 (high risk), model assigns higher risk for these patients while decreasing risks for patients with GCS=3,4,5.

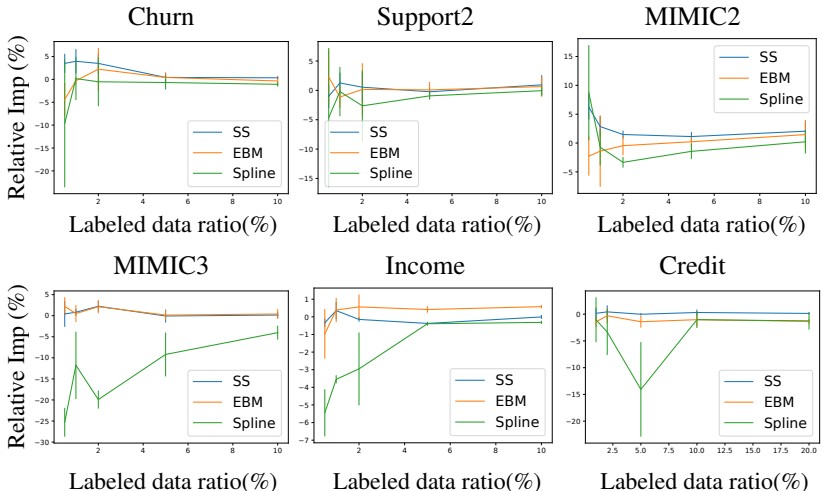

Figure 6: The relative improvement (%) over NODE-GAM without self-supervision (No-SS) on 6 datasets with various labeled data ratio. The higher number is better.

### 4.3 Self-supervised Pre-training

By training GAMs with neural networks, it enables self-supervised learning that learns representations from unlabeled data which improves accuracy in limited labeled data scenarios. We first learn a NODE-GAM that reconstructs input features under randomly masked inputs (we use 15% masks). Then we remove and re-initialize the last linear weight and fine-tune it on the original targets under limited labeled data. For fine-tuning, we freeze the embedding and only train the last linear weight for the first 500 steps; this helps stabilize the training. We also search smaller learning rates [5e−5, 1e−4, 3e−4, 5e−4] and choose the best model by validation set. We compare our self-supervised model (**SS**) with 3 other baselines: (1) NODE-GAM without self-supervision (**No-SS**), (2) EBM and (3) Spline. We randomly search 15 attention based AB-GAM architectures for both SS and No-SS.

In Figure 6, we show the relative improvement over the AUC of No-SS under variously labeled data ratio 0.5%, 1%, 2%, 5%, and 10% (except Credit which 0.5% has too few positive samples and thus crashes). And we run 3 different train-test split folds to derive mean and standard deviation. Here the relative improvement means improvement over No-SS baselines. First, we find NODE-GAM with self-supervision (SS, blue) outperforms No-SS in 6 of 7 datasets (except Income) with MIMIC2 having the most improvement (6%). This shows our NODE-GAM benefits from self-supervised pre-training. SS also outperforms EBM in 3 out of 6 datasets (Churn, MIMIC2 and Credit) up to 10% improvement in Churn, demonstrating the superiority of SS when labeled data is limited.

## 5 Limitations, Discussions and Conclusions

Although we interpret and explain the shape graphs in this paper, we want to emphasize that the shown patterns should be treated as an association not causation. Any claim based on the graphs requires a proper causal study to validate it.

In this paper, we assumed that the class of GAMs is interpretable and proposed a new deep learning model in this class, so our NODE-GAM is as interpretable to other GAMs. But readers might wonder if GAMs are interpretable to human. Hegselmann et al. (2020) show GAM is interpretable for doctors in a clinical user study; Tan et al. (2019) find GAM is more interpretable than the decision tree helping users discover more patterns and understand feature importance better; Kaur et al. (2020) compare GAM to post-hoc explanations (SHAP) and find that GAM significantly makes users answer questions more accurately, have higher confidence in explanations, and reduce cognitive load.

In this paper we propose a deep-learning version of GAM and GA$^2$M that automatically learn which main and pairwise interactions to focus without any two-stage training and model ensemble. Our GAM is also more accurate than traditional GAMs in both large datasets and limited-labeled settings. We hope this work can further inspire other interpretable design in the deep learning models.

## ACKNOWLEDGEMENT

We thank Alex Adam to provide valuable feedbacks and improve the writing of this paper. Resources used in preparing this research were provided, in part, by the Province of Ontario, the Government of Canada through CIFAR, and companies sponsoring the Vector Institute `www.vectorinstitute.ai/#partners`.

## ETHICS STATEMENT

A potential misuse of this tool is to claim causal relationships of what GAM models learned. For example, one might wrongfully claim that having asthma is good for pneumonia patients (Caruana et al., 2015). This is obviously a wrong conclusion predicated by the bias in the dataset.

The interpretable models are a great tool to discover potential biases hiding in the data. Especially in high-dimensional datasets when it's hard to pinpoint where the bias is, GAM provides easy visualization that allows users to confirm known biases and examine hidden biases. We believe that such models foster ethical adoption of machine learning in high-stakes settings. By providing a deep learning version of GAMs, our work enables the use of GAM models on larger datasets thus increasing GAM adoption in real-world settings.

## REPRODUCIBILITY STATEMENT

We released our code in `https://github.com/zzzace2000/nodegam` with instructions and hyperparameters to reproduce our final results. Our datasets can be automatically downloaded to the directory with train and test fold using included code. We report the details of our datasets in Supp. F, and hyperparameters for both our model and baselines in Supp. G. We list the hyperparameters corresponding to the best performance in Supp. H.

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

## A COMPARISON TO NAM (AGARWAL ET AL., 2020) AND THE UNIVARIATE NETWORK IN NID (TSANG ET AL., 2017)

First, we compare with NAM's (Agarwal et al., 2020) performance without and with the ensemble. Since NAM requires an extensive hyperparameter search and to be fair with NAM, we focus on 3 of their 5 datasets (MIMIC2, COMPAS, Credit) and use their reported best hyperparameters to compare. We find that NODE-GAM is better than NAM without ensemble consistently across 3 datasets.

Table 3: Comparison to NAM (Agarwal et al., 2020) with and without the ensemble.

|  | NODE GAM | NAM (no ensemble) | NAM (Ensembled) | EBM | Spline |
|---|---|---|---|---|---|
| COMPAS | 74.2 (0.9) | 73.8 (1.0) | 73.7 (1.0) | 74.3 (0.9) | 74.1 (0.9) |
| MIMIC-II | 83.2 (1.1) | 82.4 (1.0) | 83.0 (0.8) | 83.5 (1.1) | 82.5 (1.1) |
| Credit | 98.1 (1.1) | 97.5 (0.8) | 98.0 (0.2) | 97.4 (0.9) | 98.2 (1.1) |

As the reviewer points out, in the NID paper (Tsang et al., 2017) they also have a similar idea to NAM that trains a univariate network alongside an MLP to model the main effect. But there are two key differences between the univariate network in NID and NAM: (1) NAM proposes a new jumpy activation function - ExU - that can model quick, non-linear changes of the inputs as part of their hyperparameter selection, and (2) NAM uses multiple networks to ensemble. To be thorough, we compare with NAM that only uses normal units to resemble the univariate network in NID. We also considered whether removing the ensemble would disproportionately impact ExU activations since ExU activations are more prone to overfitting.

We show the performance in Table 4. We show the results in 2 of their 5 datasets (MIMIC-II, Credit) that find ExU perform better than normal units (in other 3 datasets normal units perform better). First, we find that after ensemble normal units perform quite similar to ExU in MIMIC2 but worse in Credit. And given that in 3 other datasets NAM already finds normal units to perform better, we think normal units and ExU probably have similar accuracy. Besides, ensemble helps improve performance much more for ExU units but not so much for normal units, since ExU is a more low-bias high-variance unit that benefits more from ensembles. In either case, their performance without ensemble is still inferior to our NODE-GAM.

Table 4: Comparison to NAM with normal units v.s. ExU units, and with and without ensemble.

|  | NAM-normal | NAM-normal (Ensembled) | NAM-ExU | NAM-ExU (Ensembled) |
|---|---|---|---|---|
| MIMIC-II | 82.7 (0.8) | 82.9 | 82.4 (1.0) | 83.0 (0.8) |
| Credit | 97.3 (0.8) | 97.4 | 97.5 (0.8) | 98.0 (0.2) |
| COMPAS | 73.8 (1.0) | 73.7 (1.0) | - | - |

Table 5: The default performance for 6 large datasets. The NODE-GA2M-Default is the model with default hyperparameter, and the Rel Diff is the relative difference of performance between default and tuned NODE-GA$^2$M. The first 3 datasets (Click, Epsilon and Higgs) are classification datasets and shown the Error Rate. The last 3 (Microsoft, Yahoo and Year) are shown in Mean Squared Error (MSE).

| | NODE GA$^2$M Default | NODE GA$^2$M | EBM GA$^2$M | Rel Diff |
|---|---|---|---|---|
| Click | 0.3332 ± 0.0001 | 0.3307 ± 0.0001 | 0.3297 ± 0.0001 | -0.4% |
| Epsilon | 0.1063 ± 0.0001 | 0.1050 ± 0.0002 | - | -0.8% |
| Higgs | 0.2656 ± 0.0003 | 0.2566 ± 0.0003 | 0.2767 ± 0.0004 | -3.7% |
| Microsoft | 0.5670 ± 0.0003 | 0.5617 ± 0.0003 | 0.5780 ± 0.0001 | -0.9% |
| Yahoo | 0.6002 ± 0.0004 | 0.5807 ± 0.0004 | 0.6032 ± 0.0005 | -3.4% |
| Year | 80.56 ± 0.22 | 79.57 ± 0.12 | 83.16 ± 0.01 | -1.1% |

## B  THE PERFORMANCE OF NODE-GA$^2$M AND TABNET WITH DEFAULT HYPERPARAMETERS

To increase the ease of use, we provide a strong default hyperparameter of the NODE-GA$^2$M. In Table 5, compared to the tuned NODE-GA$^2$M, the default hyperparmeter increases the error by 0.4%-3.7%, but still consistently outperforms EBM-GA$^2$M.

## C  PSEUDO-CODE FOR NODE-GAM

Here we provide the pseudo codes for our model in Alg. 1-4. We highlight our key changes that make NODE as a GAM in red, and the new architectures or regularization in blue. We show a single GAM decision tree in Alg. 1, a single GA$^2$M tree in Alg. 2, model algorithm in Alg. 3, and the model update in Alg. 4.

## D  PURIFICATION OF GA$^2$M

Note that GA$^2$M can have many representations that result in the same function. For example, for a prediction value $v$ associated with $x_2$, we can move $v$ to the main effect $f_2(x_2) = v$, or the interaction effect $f_{23}(x_2, \cdot) = v$ that involves $x_2$. To solve this ambiguity, we adopt "purification" (Lengerich et al., 2020) that pushes interaction effects into main effects if possible.

To purify an interaction $f_{jj'}$, we first bin continuous feature $x_j$ into at most $K$ quantile bins with $K$ unique values $x_j^1, ... x_j^K$ and for $x_{j'}$ as well. Then for every $x_j^k$, we move the average $a_j^k$ of interactions $f_{jj'}$ to main effects $f_j$:

$$\forall_{k=1}^K x_j^k, \quad a_j^k = \frac{1}{N_{K'}} \sum_{k'=1}^{K'} f_{jj'}(x_j^k, x_{j'}^{k'}), \quad f_{jj'}(x_j^k, x_{j'}^{k'}) = f_{jj'}(x_j^k, x_{j'}^{k'}) - a_j^k, \quad f_j(x_j^k) = f_j(x_j^k) + a_j^k$$

---

**Algorithm 1** A GAM differentiable oblivious decision tree (ODT)

---

**Input:** Input $\boldsymbol{X} \in \mathbb{R}^D$, Temperature $T$ ($T \to 0$), Previous layers' outputs $\boldsymbol{X}_p \in \mathbb{R}^P$, Previous layers' feature selection $\boldsymbol{G}_p \in \mathbb{R}^{P \times D}$, Attention Matrix $\boldsymbol{A} \in \mathbb{R}^P$

**Hyperparameters:** Tree Depth $C$, Column subsample ratio $\eta$

**Trainable Parameters:** Feature Selection Logits $\boldsymbol{F} \in \mathbb{R}^D$, Split Thresholds $\boldsymbol{b} \in \mathbb{R}^C$, Split slope $\boldsymbol{S} \in \mathbb{R}^C$, Node weights $\boldsymbol{W} \in \mathbb{R}^{2^C}$

---

**if** $\eta < 1$ **then**
    n $= \max(D\eta, 1)$         ▷ Number of subset features
    i $=$ shuffle(range(D))[int(n):]     ▷ Randomly choose features to exclude
    $\boldsymbol{F}$[i] $= -\inf$         ▷ Exclude features
**end if**
$\boldsymbol{G} = \text{EntMax}(\boldsymbol{F}/T)$     ▷ Go through EntMax to generate soft one-hot vector (Eq. 5)
$K = \boldsymbol{X} \cdot \boldsymbol{G}$     ▷ Pick 1 feature softly
**if** $\boldsymbol{X}_P$ is not None **then**
    $\boldsymbol{g} = \boldsymbol{G}_P \boldsymbol{G} \in \mathbb{R}^P$     ▷ Calculate gating weights $\boldsymbol{g}$ (Eq. 6)
    $\boldsymbol{g}' = \boldsymbol{g}/\sum \boldsymbol{g}$ **if** $\boldsymbol{A}$ is None **else** $\boldsymbol{g} \cdot \text{entmax}_\alpha(\log(\boldsymbol{g}) + \boldsymbol{A})$   ▷ Attention-based GAM (Eq.8)
    $K = K + \boldsymbol{X}_p \cdot \boldsymbol{g}'$     ▷ Add previous outputs with $\boldsymbol{g}$ normalized to 1
**end if**
$\boldsymbol{H} = \text{EntMoid}((K - \boldsymbol{b})/\boldsymbol{S}) \in \mathbb{R}^C$     ▷ Generate soft binary value
$\boldsymbol{e} = \left( \begin{bmatrix} H^1 \\ (1 - H^1) \end{bmatrix} \otimes \cdots \otimes \begin{bmatrix} (H^C) \\ (1 - H^C) \end{bmatrix} \right) \in \mathbb{R}^{2^C}$     ▷ Go through the decision tree
$h = \boldsymbol{e} \cdot \boldsymbol{W}$     ▷ Select one weight value softly as the output

**Return:** $h, \boldsymbol{G}$     ▷ Return tree response $h$ and feature selection $\boldsymbol{G}$

---

**Algorithm 2** A GA$^2$M differentiable oblivious decision tree (ODT)

---

**Input:** Input $\boldsymbol{X} \in \mathbb{R}^D$, Temperature $T$ ($T \to 0$), Previous layers' outputs $\boldsymbol{X}_p \in \mathbb{R}^P$, Previous layers' feature selection $\boldsymbol{G}_p^1, \boldsymbol{G}_p^2 \in \mathbb{R}^{P \times D}$, Attention Matrix $\boldsymbol{A} \in \mathbb{R}^P$

**Hyperparameters:** Tree Depth $C$, Column subsample ratio $\eta$

**Trainable Parameters:** Feature Selection Logits $\boldsymbol{F}^1, \boldsymbol{F}^2 \in \mathbb{R}^D$, Split Thresholds $\boldsymbol{b} \in \mathbb{R}^C$, Split slope $\boldsymbol{S} \in \mathbb{R}^C$, Node weights $\boldsymbol{W} \in \mathbb{R}^{2^C}$

---

**if** $\eta < 1$ and first time running **then**
    n $= \max(D\eta, 1)$     ▷ Number of subset features
    i $=$ shuffle(range(D))[n:]     ▷ Randomly exclude features
    $\boldsymbol{F}^1$[i] $= -\inf$, $\boldsymbol{F}^2$[i] $= -\inf$     ▷ Exclude features
**end if**
$\boldsymbol{G}^1 = \text{EntMax}(\boldsymbol{F}^1/T)$, $\boldsymbol{G}^2 = \text{EntMax}(\boldsymbol{F}^2/T)$     ▷ Get soft one-hot vector (Eq. 5)
$K^1 = \boldsymbol{X} \cdot \boldsymbol{G}^1$, $K^2 = \boldsymbol{X} \cdot \boldsymbol{G}^2$     ▷ Pick 1 feature softly
**if** $\boldsymbol{X}_P$ is not None **then**
    $\boldsymbol{g} = min((\boldsymbol{G}^1 \cdot \boldsymbol{G}_P^1) \times (\boldsymbol{G}^2 \cdot \boldsymbol{G}_P^2) + (\boldsymbol{G}^1 \cdot \boldsymbol{G}_P^2) \times (\boldsymbol{G}^2 \cdot \boldsymbol{G}_P^1), 1)$ ▷ Gating weights (Eq. 10)
    $\boldsymbol{g}' = \boldsymbol{g}/\sum \boldsymbol{g}$ **if** $\boldsymbol{A}$ is None **else** $\boldsymbol{g} \cdot \text{entmax}_\alpha(\log(\boldsymbol{g}) + \boldsymbol{A})$   ▷ Attention-based GAM (Eq.8)
    $K^1 = K^1 + \boldsymbol{X}_p \cdot \boldsymbol{g}'$     ▷ Add previous outputs with $\boldsymbol{g}$ normalized to 1
    $K^2 = K^2 + \boldsymbol{X}_p \cdot \boldsymbol{g}'$
    $\boldsymbol{K} = [K^1, K^2, K^1, ..., K^2] \in \mathbb{R}^C$     ▷ Alternating between $K^1$ and $K^2$
**end if**
$\boldsymbol{H} = \text{EntMoid}((\boldsymbol{K} - \boldsymbol{b})/\boldsymbol{S}) \in \mathbb{R}^C$     ▷ Generate soft binary value
$\boldsymbol{e} = \left( \begin{bmatrix} H^1 \\ (1 - H^1) \end{bmatrix} \otimes \cdots \otimes \begin{bmatrix} (H^C) \\ (1 - H^C) \end{bmatrix} \right) \in \mathbb{R}^{2^C}$     ▷ Go through the decision tree
$h = \boldsymbol{e} \cdot \boldsymbol{W}$     ▷ Select one weight value softly as the output

**Return:** $h, [\boldsymbol{G}^1, \boldsymbol{G}^2]$     ▷ Return tree response $h$ and feature selection $[\boldsymbol{G}^1, \boldsymbol{G}^2]$

---

This is one step to purify $f_{jj'}$ to $f_j$. Then we purify $f_{jj'}$ to $f_{j'}$, and so on until all $a_j^k$ and $a_{j'}^{k'}$ are close to 0.

---

**Algorithm 3** The NODE-GAM / NODE-GA$^2$M algorithm

---

1: **Input:** Input $\boldsymbol{X} \in \mathbb{R}^D$
2: **Hyperparameters:** number of layers $L$, number of trees $I$ per layer, tree depth $C$, current optimization step $s$, temperature annealing step $S$, Attention Embedding $E$
3: **Trainable Parameters**: the decision trees $\boldsymbol{M}_l$ in each layer $l$ (either GAM trees (Alg. 1) or GA$^2$M trees (Alg. 2)), the final output weights $\boldsymbol{W}_L \in \mathbb{R}^{(LI)}$ and bias $w_0$
4:

---

5: **if** $E > 0$ **then**                     ▷ Use attention-based GAM
6:   Initialize $\boldsymbol{B}^l \in \mathbb{R}^{(l-1)I \times E}$ and $\boldsymbol{C}^l \in \mathbb{R}^{E \times I}$ for $l = 2...L$
7: **end if**
8: $T = 10^{-2(s/S)}$ **if** $s \leq S$ **else** $0$            ▷ Slowly decrease temperature to $0$
9: $\boldsymbol{X}_p = $ None, $\boldsymbol{G}_p = $ None    ▷ Initialize previous trees' outputs $\boldsymbol{X}_p$ and feature selections $\boldsymbol{G}_p$
10: **for** $l = 1$ **to** $L$ **do**
11:   $\boldsymbol{A}^l = $ None **if** $E = 0$ **or** $l = 1$ **else** $\boldsymbol{B}^l\boldsymbol{C}^l$         ▷ Calculate attention matrix
12:   $\boldsymbol{h}_l, \boldsymbol{G}_l = \boldsymbol{M}_l(\boldsymbol{X}, T, \boldsymbol{X}_p, \boldsymbol{G}_p, \boldsymbol{A}^l)$     ▷ Run total $I$ trees in $\boldsymbol{M}_l$ by Alg. 1 or 2
13:   $\boldsymbol{h}_l = $ Dropout$(\boldsymbol{h}_l)$                   ▷ Dropout rate $p_1$
14:   $\boldsymbol{X}_p = \boldsymbol{h}_l$ **if** $\boldsymbol{X}_p$ is None **else** $[\boldsymbol{X}_p, \boldsymbol{h}_l]$       ▷ Concatenate outputs $\boldsymbol{h}_l$
15:   $\boldsymbol{G}_p = \boldsymbol{G}_l$ **if** $\boldsymbol{G}_p$ is None **else** $[\boldsymbol{G}_p, \boldsymbol{G}_l]$     ▷ Concatenate feature selection $\boldsymbol{G}_l$
16: **end for**
17: $\boldsymbol{W}_L = $ Dropout$(\boldsymbol{W}_L)$                  ▷ Dropout rate $p_2$
18: $R = \boldsymbol{X}_p \cdot \boldsymbol{W}_L + w_0$             ▷ Go through last linear layer
19: **Return:** $R, \boldsymbol{X}_P$       ▷ Return model response $R$ and all trees' outputs $\boldsymbol{X}_P$

---

**Algorithm 4** Our model's update

---

1:
2: **Input:** An input $\boldsymbol{X} \in \mathbb{R}^D$, target $y$, Node-GAM model $\mathcal{M}$

---

3: $R, \boldsymbol{X}_P = \mathcal{M}(\boldsymbol{X})$                ▷ Run Node-GAM (Alg. 3)
4: $L = $ BCELoss$(y, R)$ **if** binary classification **else** MSELoss$(y, R)$
5: $L = L + \lambda \sum(\boldsymbol{X}_P)^2$           ▷ Add the $\ell_2$ on the output of trees
6: Optimize $L$ by Adam optimizer

---

## E  NODE-GA$^2$M FIGURES

Here we show the architecutre of NODE-GA$^2$M in Figure 7.

## F  DATASET DESCRIPTIONS

Here we describe all $8$ datasets we use and we summarize them in Table 6.

- Churn: this is to predict which user is a potential subscription churner for telecom company. `https://www.kaggle.com/blastchar/telco-customer-churn`

- Support2: this is to predict mortality in the hospital by several lab values. `http://biostat.mc.vanderbilt.edu/DataSets`

- MIMIC-II and MIMIC-III dataset (Johnson et al., 2016b): this is an ICU patient datasets to predict mortality of patients in a tertiary academic medical center in Boston, MA, USA.

- Income: UCI Dua & Graff (2017). This is a dataset from census collected in 1994, and the goal is to predict who has income >50K/year. `https://archive.ics.uci.edu/ml/datasets/adult`

- Credit: this is to predict which transaction is a fraud. The features provided are the coefficient of PCA components to protect privacy. `https://www.kaggle.com/mlg-ulb/creditcardfraud`

- Bikeshare (Dua & Graff, 2017): this is the hourly bikeshare rental counts in Washington D.C., USA. `https://archive.ics.uci.edu/ml/datasets/bike+sharing+dataset`

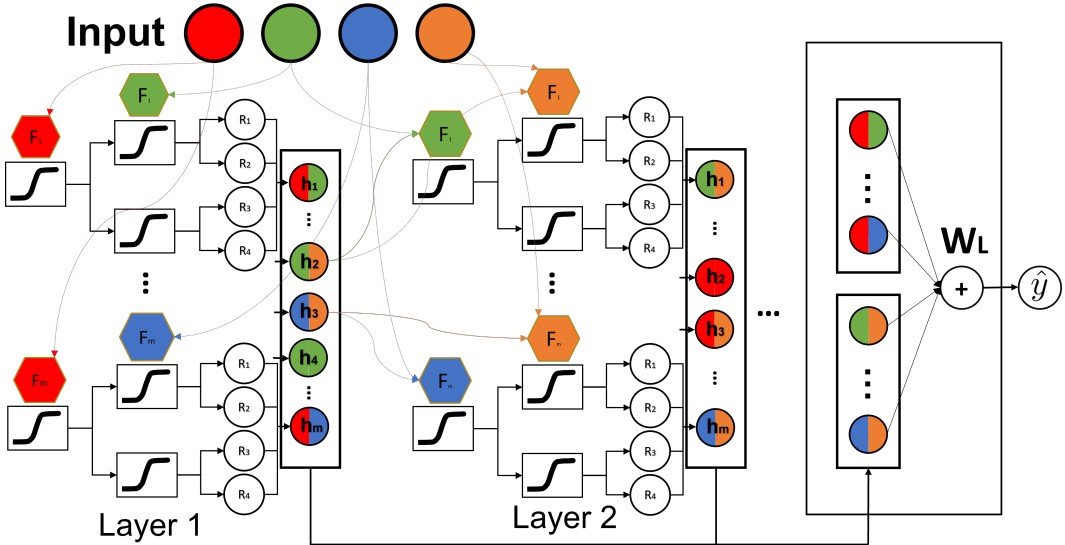

Figure 7: The NODE-GA$^2$M architecture. Here we show 4 features with 4 different colors. Each layer consists of $I$ differentiable oblivious decision trees that outputs $h_1...h_I$, where each $h_i$ depends on at most 2 features. We only connect trees between layers if two trees depend on the same two features. And we concatenate all outputs from all layers as inputs to the last linear layer $\boldsymbol{W}_L$ to produce outputs.

- Wine (Dua & Graff, 2017): this is to predict the wine quality based on a variety of lab values. https://archive.ics.uci.edu/ml/datasets/wine+quality

For 6 datasets used in NODE, we use the scripts from NODE paper (https://github.com/Qwicen/node) which directly downloads the dataset. Here we still cite and list their sources:

- Click: https://www.kaggle.com/c/kddcup2012-track2

- Higgs: UCI (Dua & Graff, 2017) https://archive.ics.uci.edu/ml/datasets/HIGGS

- Epsilon: https://www.k4all.org/project/large-scale-learning-challenge/

- Microsoft: https://www.microsoft.com/en-us/research/project/mslr/

- Yahoo: https://webscope.sandbox.yahoo.com/catalog.php?datatype=c.

- Year (Dua & Graff, 2017): https://archive.ics.uci.edu/ml/datasets/yearpredictionmsd

## F.1 PREPROCESSING

For NODE and NODE-GAM/GA$^2$M, we follow Popov et al. (2019) to do target encoding for categorical features, and do quantile transform[1] with 2000 bins for all features to Gaussian distribution (we find Gaussian performs better than Uniform). We find adding small gaussian noise (e.g. 1e-5) when fitting quantile transformation (but no noise in transformation stage) is crucial to have mean 0 and standard deviation close to 1 after transformation.

---

[1] sklearn.preprocessing.quantile_transform

Table 6: All dataset statistics and descriptions.

| | Domain | # Samples | # Features | Positive rate | Description |
|---|---|---|---|---|---|
| Churn | Retail | 7,043 | 19 | 26.54% | Subscription churner |
| Support2 | Healthcare | 9,105 | 29 | 25.92% | Hospital mortality |
| MIMIC-II | Healthcare | 24,508 | 17 | 12.25% | ICU mortality |
| MIMIC-III | Healthcare | 27,348 | 57 | 9.84% | ICU mortality |
| Income | Finance | 32,561 | 14 | 24.08% | Income prediction |
| Credit | Retail | 284,807 | 30 | 0.17% | Fraud detection |
| Bikeshare | Retail | 17,389 | 16 | - | Bikeshare rental counts |
| Wine | Nature | 4,898 | 12 | - | Wine quality |
| Click | Ads | 1M | 11 | 50% | 2012 KDD Cup |
| Higgs | Nature | 11M | 28 | 53% | Higgs bosons prediction |
| Epsilon | - | 500K | 2k | 50% | PASCAL Challenge 2008 |
| Microsoft | Ads | 964K | 136 | - | MSLR-WEB10K |
| Yahoo | Ads | 709K | 699 | - | Yahoo LETOR dataset |
| Year | Music | 515K | 90 | - | Million Song Dataset |

## G  HYPERPARAMETERS SELECTION

In order to tune the hyperparameters, we performed a random stratified split of full training data into train set (80%) and validation set (20%) for all datasets. For datasets we compile of medium-sized (Income, Churn, Credit, Mimic2, Mimic3, Support2, Bikeshare), we do a 5-fold cross validation for 5 different test splits. For datasets in NODE paper (Click, Epsilon, Higgs, Microsoft, Yahoo, Year), we use train/val/test split provided by the NODE paper author. Since they only provide 1 test split, we report standard deviation by different random seeds on these datasets. For medium-sized datasets, we only tune hyperparameters on the first train-val-test fold split, and fix the hyperparameters to run the rest of 4 folds. This means that we do not search hyperparameters per fold to avoid computational overheads. All NODE, NODE-GAM/GA$^2$M are run with 1 TITAN Xp GPU, 4 CPU and 8GB memory. For EBM and Spline, they are run with a machine with 32 CPUs and 120GB memory.

Below we describe the hyperparameters we use for each method:

### G.1  EBM

For EBM, we set inner_bags=100 and outer_bags=100 and set the maximum rounds as 20k to make sure it converges; we find EBM performs very stable out of this choice probably because we set total bagging as 10k that makes it stable; other parameters have little effect on final performance.

For EBM GA$^2$M, we search the number of interactions for 16, 32, 64, 128 and choose the best one on validation set. On large datasets we set the number of iterations as 64 as we find it performs quite well on medium-sized datasets.

### G.2  SPLINE

We use the cubic spline in PyGAM package (Servén & Brummitt, 2018) that we follow Chang et al. (2021) to set the number of knots per feature to a large number 50 (we find setting it larger would crash the model), and search the best lambda penalty between 1e-3 to 1e3 for 15 times and return the best model.

### G.3  NODE, NODE-GA$^2$M AND NODE

We follow NODE to use QHAdam (Ma & Yarats, 2018) and average the most recent 5 checkpoints. In addition, we adopt learning rate warmup at first 500 steps. And we early stop our training for no improvement for 11k steps and decay learning rate to 1/5 if no improvement happens in 5k steps.

Here we list the hyperparameters we find works quite well and we do not do random search on these hyperparameters:

- *optimizer*: QHAdam (Ma & Yarats, 2018) (same as NODE paper)
- *lr_warmup_steps*: 500
- *num_checkpoints_avged*: 5
- *temperature_annealing_steps* ($S$): 4k
- *min_temperature*: 0.01 (0.01 is small enough for making one-hot vector. And after $S$ steps we set the function to produce one-hot vector exactly.)
- *batch_size*: 2048, or the max batch size that fits in GPU memory with minimum 128.
- *Maximum training time*: 20 hours. This is just to avoid model training for too long.

We use random search to find the best hyperparameters which we list the range in below. We list the random search range for NODE:

- *num_layers*: {2, 3, 4, 5}. Default: 3.
- *total tree counts* (= *num_trees* × *num_layers*): {500, 1000, 2000, 4000}. Default: 2000.
- *depth*: {2, 4, 6}. Default: 4.
- *tree_dim*: {0, 1}. Default: 0.
- *output_dropout* ($p1$): {0, 0.1, 0.2}. Default: 0.
- *colsample_bytree*: {1, 0.5, 0.1, 1e-5}. Default: 0.1.
- *lr*: {0.01, 0.005}. Default: 0.01.
- *l2_lambda*: {0., 1e-7, 1e-6, 1e-5}. Default: 1e-5.
- *add_last_linear* (to add last linear weight or not): {0, 1}. Default: 1.
- *last_dropout* ($p2$, only if *add_last_linear*=1): {0, 0.1, 0.2, 0.3}. Default: 0.5.
- *seed*: uniform distribution [1, 100].

For NODE-GAM and NODE-GA$^2$M, we have additional parameters:

- *arch*: {GAM, AB-GAM}. Default: AB-GAM.
- *dim_att* (dimension of attention embedding $E$): {8, 16, 32}. Default: 16.

We show the best hyperparameters for each dataset in Section H. And we show the performance of default hyperparameter in Suppl. B,

### G.4 XGBoost

For large datasets in NODE, we directly report the performance from the original NODE paper. For medium-sized data, we set the depth of xgboost as 3, and learning rate as 0.1 with *n_estimators*=50k and set early stopping for 50 rounds to make sure it converges.

### G.5 Random Forest (RF)

We use the default hyperparameters from sklearn and set the number of trees to a large number 1000.

## H Best hyperparameters found in each dataset

Here we report the best hyperparameters we find for 9 medium-sized datasets in Table 7 (NODE-GAM), Table 8 (NODE-GA$^2$M), and Table 9 (NODE). We report the best hyperparameters for large datasets in Table 10 (NODE-GAM) and Table 11 (NODE-GA$^2$M).

Table 7: The best hyperparameters for NODE-GAM architecture.

| Dataset | Compas | Churn | Support2 | Mimic2 | Mimic3 | Adult | Credit | Bikeshare | Wine |
|---|---|---|---|---|---|---|---|---|---|
| batch size | 2048 | 2048 | 2048 | 2048 | 512 | 2048 | 2048 | 2048 | 2048 |
| num layers | 5 | 3 | 4 | 4 | 3 | 3 | 5 | 2 | 5 |
| num trees | 800 | 166 | 125 | 500 | 1333 | 666 | 400 | 250 | 800 |
| depth | 4 | 4 | 2 | 4 | 6 | 4 | 2 | 2 | 2 |
| addi tree dim | 2 | 2 | 1 | 1 | 0 | 1 | 2 | 1 | 1 |
| output dropout | 0.3 | 0.1 | 0.1 | 0 | 0.2 | 0.1 | 0.2 | 0.2 | 0 |
| colsample bytree | 0.5 | 0.5 | 1e-5 | 0.5 | 1e-5 | 0.5 | 0.1 | 0.5 | 0.5 |
| lr | 0.01 | 0.005 | 0.01 | 0.01 | 0.005 | 0.01 | 0.01 | 0.005 | 0.005 |
| l2 lambda | 1e-5 | 1e-5 | 1e-6 | 1e-7 | 1e-7 | 0 | 0 | 1e-7 | 1e-5 |
| add last linear | 1 | 1 | 1 | 0 | 1 | 1 | 1 | 1 | 1 |
| last dropout | 0 | 0 | 0 | 0 | 0 | 0 | 0 | 0.3 | 0.1 |
| seed | 67 | 48 | 43 | 99 | 97 | 46 | 87 | 55 | 31 |
| arch | AB-GAM | AB-GAM | GAM | AB-GAM | GAM | GAM | AB-GAM | GAM | GAM |
| dim att | 16 | 8 | - | 32 | - | - | 8 | - | - |

## I    COMPLETE SHAPE GRAPHS IN BIKESHARE AND MIMIC2

We list all main effects of Bikeshare in Fig. 8 and top 16 interactions effects in Fig. 9. We list all main effects of MIMIC2 in Fig. 10 and top 16 interactions effects in Fig. 11.

Table 8: The best hyperparameters for NODE-GA$^2$M architecture.

| Dataset | Compas | Churn | Support2 | Mimic2 | Mimic3 | Adult | Credit | Bikeshare | Wine |
|---|---|---|---|---|---|---|---|---|---|
| batch size | 2048 | 2048 | 256 | 256 | 512 | 256 | 512 | 2048 | 512 |
| num layers | 4 | 3 | 2 | 2 | 4 | 2 | 2 | 4 | 4 |
| num trees | 1000 | 333 | 2000 | 2000 | 1000 | 2000 | 1000 | 125 | 1000 |
| depth | 2 | 2 | 6 | 6 | 6 | 6 | 6 | 6 | 6 |
| addi tree dim | 2 | 2 | 2 | 0 | 1 | 2 | 0 | 1 | 1 |
| output dropout | 0.2 | 0 | 0.1 | 0 | 0.2 | 0.1 | 0.2 | 0 | 0.2 |
| colsample bytree | 0.2 | 0.5 | 1 | 0.2 | 0.5 | 1 | 0.2 | 0.5 | 0.5 |
| lr | 0.005 | 0.005 | 0.01 | 0.005 | 0.01 | 0.01 | 0.01 | 0.01 | 0.01 |
| l2 lambda | 0 | 0 | 0 | 1e-5 | 0 | 0 | 0 | 0 | 0 |
| add last linear | 1 | 0 | 0 | 0 | 0 | 1 | 1 | 1 | 0 |
| last dropout | 0.2 | 0.2 | 0 | 0 | 0 | 0 | 0 | 0.3 | 0 |
| seed | 32 | 31 | 33 | 10 | 87 | 33 | 38 | 83 | 87 |
| arch | GAM | AB-GAM | AB-GAM | AB-GAM | AB-GAM | GAM | AB-GAM | GAM | AB-GAM |
| dim att | - | 32 | 32 | 8 | 16 | - | 32 | - | 16 |

Table 9: The best hyperparameters for NODE architecture.

| Dataset | Compas | Churn | Support2 | Mimic2 | Mimic3 | Adult | Credit | Bikeshare | Wine |
|---|---|---|---|---|---|---|---|---|---|
| batch size | 2048 | 2048 | 2048 | 2048 | 2048 | 2048 | 512 | 2048 | 2048 |
| num layers | 5 | 4 | 2 | 3 | 2 | 2 | 3 | 3 | 2 |
| num trees | 100 | 125 | 1000 | 166 | 1000 | 1000 | 1333 | 333 | 500 |
| depth | 2 | 2 | 4 | 6 | 4 | 4 | 6 | 4 | 4 |
| addi tree dim | 1 | 0 | 0 | 0 | 0 | 0 | 1 | 1 | 1 |
| output dropout | 0 | 0 | 0.2 | 0.2 | 0.2 | 0.2 | 0.2 | 0.1 | 0 |
| colsample bytree | 0.2 | 0.5 | 0.2 | 0.2 | 0.2 | 0.2 | 0.2 | 0.5 | 1 |
| lr | 0.005 | 0.005 | 0.005 | 0.005 | 0.005 | 0.005 | 0.005 | 0.005 | 0.01 |
| l2 lambda | 0 | 1e-5 | 1e-7 | 1e-6 | 1e-7 | 1e-7 | 1e-6 | 1e-5 | 0 |
| add last linear | 0 | 0 | 0 | 0 | 0 | 0 | 1 | 1 | 0 |
| last dropout | 0 | 0 | 0 | 0 | 0 | 0 | 0 | 0.3 | 0 |
| seed | 3 | 26 | 93 | 17 | 93 | 93 | 82 | 49 | 73 |

Table 10: The best hyperparameters for NODE-GAM architecture for 6 large datasets.

| Dataset | Click | Epsilon | Higgs | Microsoft | Yahoo | Year |
|---|---|---|---|---|---|---|
| *batch size* | 2048 | 2048 | 2048 | 2048 | 2048 | 2048 |
| *num layers* | 5 | 5 | 5 | 4 | 4 | 2 |
| *num trees* | 800 | 400 | 200 | 125 | 500 | 500 |
| *depth* | 4 | 4 | 4 | 6 | 4 | 2 |
| *addi tree dim* | 2 | 2 | 2 | 2 | 0 | 1 |
| *output dropout* | 0 | 0.1 | 0 | 0.1 | 0.2 | 0.1 |
| *colsample bytree* | 1e-5 | 0.1 | 0.5 | 0.1 | 0.1 | 0.5 |
| *lr* | 5e-3 | 1e-2 | 5e-3 | 5e-3 | 5e-3 | 1e-2 |
| *l2 lambda* | 1e-7 | 0 | 1e-5 | 0 | 1e-6 | 1e-6 |
| *add last linear* | 0 | 1 | 1 | 0 | 0 | 1 |
| *last dropout* | 0 | 0.1 | 0 | 0.1 | 0.2 | 0.1 |
| *seed* | 97 | 31 | 67 | 67 | 14 | 58 |
| *arch* | AB-GAM | AB-GAM | AB-GAM | AB-GAM | AB-GAM | AB-GAM |
| *dim att* | 32 | 16 | 32 | 8 | 8 | 16 |

Table 11: The best hyperparameters for NODE-GA$^2$M architecture for 6 large datasets.

| Dataset | Click | Epsilon | Higgs | Microsoft | Yahoo | Year |
|---|---|---|---|---|---|---|
| *batch size* | 2048 | 2048 | 2048 | 1024 | 2048 | 512 |
| *num layers* | 3 | 2 | 2 | 4 | 5 | 5 |
| *num trees* | 1333 | 2000 | 1000 | 500 | 800 | 800 |
| *depth* | 4 | 2 | 4 | 6 | 4 | 6 |
| *addi tree dim* | 2 | 2 | 0 | 0 | 0 | 0 |
| *output dropout* | 0.2 | 0.2 | 0 | 0.1 | 0.2 | 0.2 |
| *colsample bytree* | 0.5 | 0.5 | 1 | 1 | 0.5 | 1 |
| *lr* | 0.005 | 0.01 | 0.01 | 0.005 | 0.005 | 0.005 |
| *l2 lambda* | 1e-6 | 1e-6 | 1e-6 | 0 | 0 | 1e-6 |
| *add last linear* | 1 | 1 | 1 | 1 | 1 | 1 |
| *last dropout* | 0.15 | 0.3 | 0 | 0.15 | 0 | 0 |
| *seed* | 36 | 5 | 95 | 69 | 25 | 78 |
| *arch* | AB-GAM | AB-GAM | AB-GAM | AB-GAM | AB-GAM | AB-GAM |
| *dim att* | 32 | 32 | 8 | 8 | 32 | 16 |

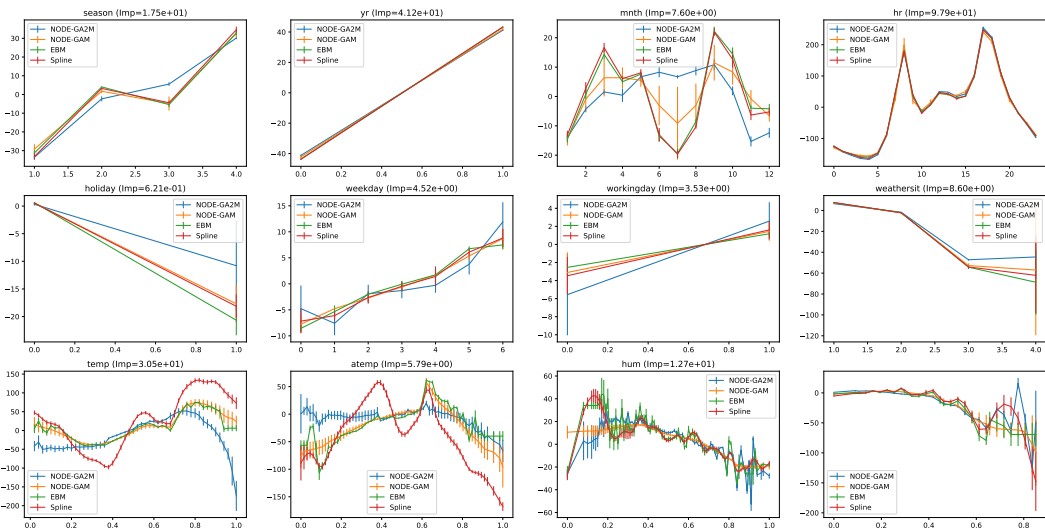

Figure 8: The shape plots of all features (main effects) in Bikeshare. We also show the feature importance (Imp).

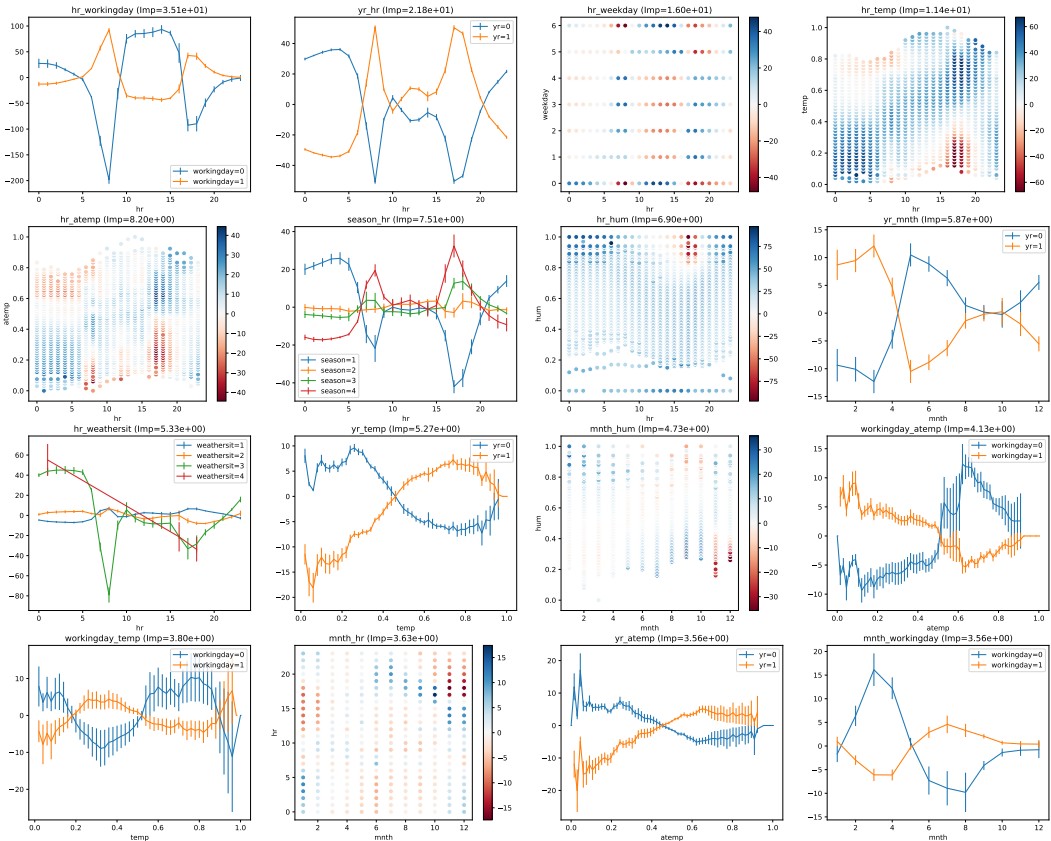

Figure 9: The shape plots of top 16 interactions in Bikeshare. We also show the feature importance (Imp).

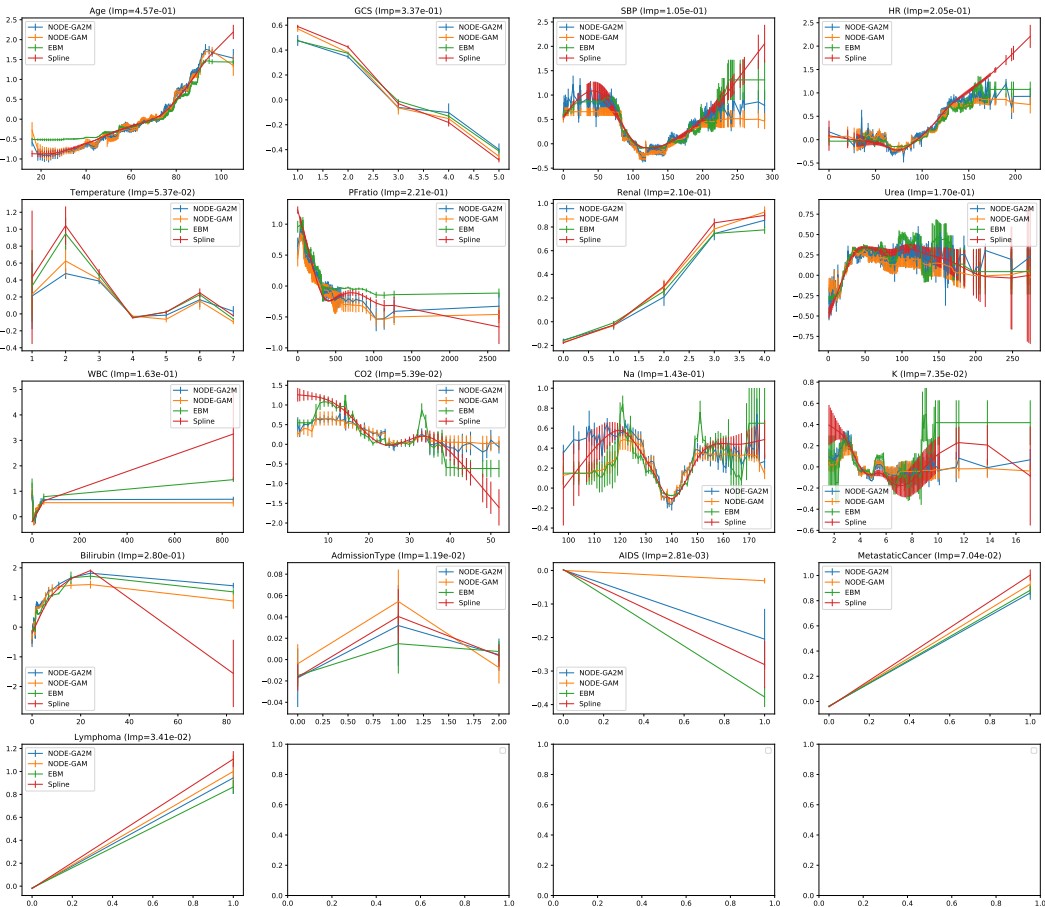

Figure 10: The shape plots of all features (main effects) in MIMIC2. We also show the feature importance (Imp).

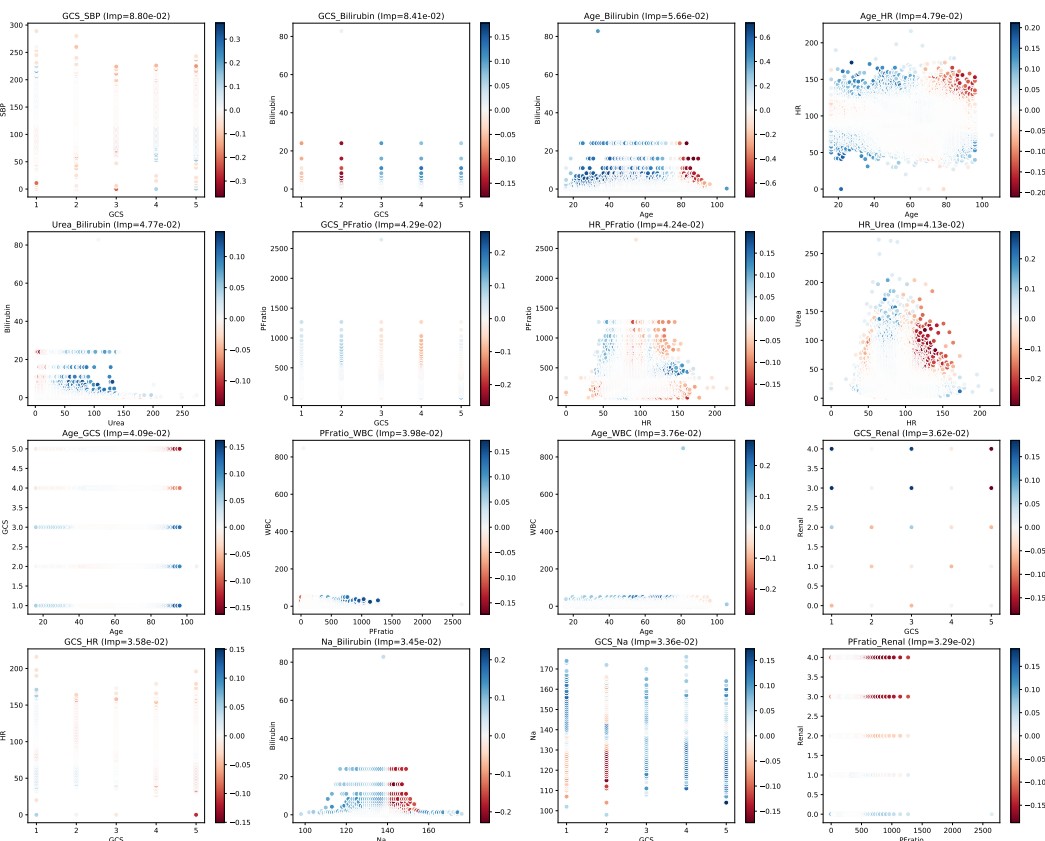

Figure 11: The shape plots of top 16 interactions in MIMIC2. We also show the feature importance (Imp).

