# OpenReview forum: "NODE-GAM: Neural Generalized Additive Model for Interpretable Deep Learning"
_ICLR.cc/2022/Conference — ICLR 2022 Spotlight_

### Official Review · Reviewer_qoW7 · 2021-11-02

**Correctness:** 4
**Technical Novelty And Significance:** 3
**Empirical Novelty And Significance:** Not applicable
**Recommendation:** 8
**Confidence:** 3

**Main Review:**

Pros:
- I believe that there is well backed motivation for work based off of the plentiful literature review.
- There is novel integration of 2 methods previously not combined before
- A variety of datasets seem to show that the proposed method is useful.
- Code is available for reproducibility.

Cons: The only issues I have with this are things that could improve clarity.
- If it is possible, I would love to see a full algorithm of NODE-GAM for clarity and understanding of each novel step
- I would rather the the argument for the interpretability be moved to the introduction / literature review section rather than in the conclusion for flow. Then, the authors could summarize their work in the conclusion instead.

**Summary Of The Paper:**

Generalized Additive Models (GAMs) are a class of interpretable models with a long history of use in these high-risk domains, but they lack desirable features of deep learning such as differentiability and scalability.

The authors propose a neural GAM (NODE-GAM) and neural GA2M (NODE-GA2M) that scale well and perform better than other GAMs on large datasets, while remaining interpretable.

Popov et al. (2019) developed NODE that mimics an ensemble of decision trees, but suffers from lack of interpretability similarly to other ensemble and deep learning models. Agarwal et al. (2020) developed a Neural Additive Model (NAM) whose deep learning architecture is a GAM, but does not scale well.

The authors combine the above mentioned 2 approaches via a 3 step process:
1. Instead of letting the ODT feature function be a weighted sum of features, they make it pick only one feature.
2. Within each tree, they make a logit layer the same across all depths to remove interactions between features.
3. Finally, they avoid the DenseNet connection between two trees that focus on different features.

The authors then output performance on 6 popular binary classification datasets (Churn, Support2, MIMIC2, MIMIC3, Income, and Credit) and 2 regression datasets (Wine and Bikeshare). Looking at the experiments, their NODE GA^2M version seems to perform similar to the compared state of the art.

The authors additionally perform an ablation experiment by testing their method's semi supervised learning capabilities, show that it is able to learn on mask data and be fine tuned on a limited labeled dataset.

Finally, the conclusion has arguments to support the interpretability of GAMs and to defend their model's interpretability

**Summary Of The Review:**

I would tend to accept this paper as it is novel enough and supported by empirical experiments.

---

> ### Author Response · Authors · 2021-11-16
> **Thank you for tending to accept this paper**
>
> > If it is possible, I would love to see a full algorithm of NODE-GAM for clarity and understanding of each novel step
>
> We add a complete pseudo-code in Supp. B. Also, in the process of writing this, we realize we forget to mention the hyperparameters subsampling (basically only allow each tree to model on a fraction of features like Random Forest). We add it and sincerely thank you for this advice. Please let us know if there's anything still unclear.
>
> > I would rather the the argument for the interpretability be moved to the introduction / literature review section rather than in the conclusion for flow. Then, the authors could summarize their work in the conclusion instead.
>
> That's a good suggestion. But we feel moving it to the intro also hinders the flow. So in light of this, we shorten that paragraph and add another conclusion paragraph in the end, and hopefully that will read much better (at least the last paragraph is the conclusion). Let us know if you like it!

---

> > ### Comment · Reviewer_qoW7 · 2021-11-22
> > **I now firmly feel that this paper should be accepted.**
> >
> > I have changed my score to reflect it as such!

---

> > > ### Author Response · Authors · 2021-11-22
> > > **Thank you!**
> > >
> > > Thank you and it makes my day :)

---

### Official Review · Reviewer_5LN4 · 2021-11-02

**Correctness:** 2
**Technical Novelty And Significance:** 3
**Empirical Novelty And Significance:** 2
**Recommendation:** 5
**Confidence:** 2

**Main Review:**

For the design of the new architecture, the authors made three major changes to the NODE architectures. However, it was not very clear to me the considerations behind, e.g. why avoiding features interactions in a tree. It will be great if the authors can provide evidence, either theoretical or empirical, to support the suggested designs.

**Summary Of The Paper:**

The authors proposed novel architectures for neural GAM and GA2M, which preserves the interpretability of GAM and leveraging the deep learning architectures for performance gains. The new method was assessed on 14 different dataset which covers a wide variety of prediction tasks. The performance of methods (NODE-GAM and NODE-GA2M) were comparable with the other GAM methods. Compared to other similar methods, the proposed method has a better scale ability, and better performance on larger datasets.

**Summary Of The Review:**

The paper proposed an interesting architecture for interpretable deep learning models. However, based on the empirical results, the method does not seem to be superior or only marginally better than traditional methods such as the spline model.

---

> ### Author Response · Authors · 2021-11-16
> **Thank you for your review**
>
> > For the design of the new architecture, the authors made three major changes to the NODE architectures. However, it was not very clear to me the considerations behind, e.g. why avoiding features interactions in a tree. It will be great if the authors can provide evidence, either theoretical or empirical, to support the suggested designs.
>
> We thank you for being honest that you don't understand the design and the intuition behind it.
>
> The intuition is to make a deep learning model not use all the interactions among features, but only use up to the 2nd order interactions (i.e. pairwise interactions). By doing this, the whole model can be expressed as an addition of bias, main, and pairwise terms (Sec. 2) which enables us to visualize each individual term as a graph. Humans can read it off the figure to understand how model makes predictions.
>
> One example of the findings (Fig. 3(b)) is that people often rent more bikes at 9 AM and 5 PM on Monday to Thursday, which could be explained by people renting bikes to commute. But on Friday people rent bikes around 10 AM and 4 PM. This probably shows people go to work a bit late and get off work earlier on Friday. These insights are very hard for other models to gain like deep neural networks, even by using post-hoc explanations like SHAP. We hope it makes more sense now!
>
>
> > The paper proposed an interesting architecture for interpretable deep learning models. However, based on the empirical results, the method does not seem to be superior or only marginally better than traditional methods such as the spline model.
>
> We hope our improvement over other GAMs in larger datasets (Table 2) up to 7% and self-supervised learning (Figure 6) up to 10% can alleviate your concern!

---

### Official Review · Reviewer_L13g · 2021-11-06

**Correctness:** 4
**Technical Novelty And Significance:** 3
**Empirical Novelty And Significance:** 3
**Recommendation:** 8
**Confidence:** 4

**Main Review:**

The authors added a comparison to NAM without bagging and provided more context for the advantages of their approach relative to other works. As a result, I am increasing my score.

----

Strengths
- The authors put a lot of thought into designing the architectures of NODE-GAM and NODE-GA$^2$M. I appreciated the clever tricks adopted, like the gating mechanism and the annealing to constrain to 1 feature, as well as the small extensions to NODE like adding attention and regularization.
- The authors similarly put careful thought into extracting the interactions. I especially appreciate the application of the "purification" technique (Lengerich et al. 2020), which addresses an ambiguity problem that has bothered me about approaches like these in the past.

Weaknesses
- The biggest weakness for me is primarily some missing baselines that I would be interested in seeing. In reading this paper, I felt there was an "elephant in the room" question lurking in my mind about why a much simpler neural-network-based architecture, such as the "univariate" networks present in the "Neural Interaction Detection" (NID) paper for modeling main effects (Fig 2 in https://arxiv.org/pdf/1705.04977.pdf; not cited) was not applied. I believe this is similar to the approach taken in the NAM paper as well (https://arxiv.org/pdf/2004.13912.pdf; cited). Regarding the NAM paper, the authors write "NAM requires training on tens to hundreds of neural networks and requires an extensive hyperparameter search...in addition, NAM's shape graphs are too smooth after bagging". As a reader, having seen the "univariate networks" used in the NID paper (which, to my knowledge, were not trained with tens to hundreds of networks), I'm left wondering "what happens if you use fewer neural networks and don't use bagging"? I'm sure other difficulties arise in this case - perhaps the model used in the NID paper is also not good at modeling abrupt changes - but it would be helpful to the reader to showcase to what extent this is an issue in practice. I get the sense that the authors are familiar with a lot of not-well-documented limitations of other approaches that led them to develop the NODE-GAM architecture, and it would be helpful if these limitations were made explicit.
- Still on the subject of prior work/baselines, I am interested to hear how this approach compares to the "Neural Interaction Transparency" (NIT) approach (https://proceedings.neurips.cc/paper/2018/hash/74378afe5e8b20910cf1f939e57f0480-Abstract.html) - it does not seem as though NIT is cited in this work. To quote from the abstract of NIT: "NIT is also flexible and efficient; it can learn generalized additive models with maximum $K$-order interactions by training only $O(1)$ models" - so it seems quite topical.

Minor
- There are a few very noticeable typos scattered around (e.g. "Mathamatically" at the bottom of page 2, "distirbution" at the top of page 5).
- In equation 7, it looks like the attention weights $a_{\hat{i}i}$ are not indexed by the layer $l$ - is this just missing notation? Presumably the attention weights vary depending on the layer?
- I was a bit confused by the sentence "To allow two-way interactions, for each tree we introduce two logits $\boldsymbol{F}^1$ and $\boldsymbol{F}^2$ instead of just one, and let $\boldsymbol{F}^c = \boldsymbol{F}^{\lfloor c/2 \rfloor }$ for $c > 2$". Maybe the authors meant to write $\boldsymbol{F}^c = \boldsymbol{F}^{(c \mod 2) + 1}$ so that $\boldsymbol{F}^c$ alternates between $\boldsymbol{F}^1$ and $\boldsymbol{F}^2$? $\lfloor c/2 \rfloor$ can take on values beyond 1 and 2.
- In the section on regularization, the authors mention that adding a constant corresponding to the log of the class imbalance is needed for the $l_2$ penalty to work because "$l_2$ induces the model output to 0". Wouldn't it be easier to solve this by adding an (unregularized) learnable bias term to the output?


**Summary Of The Paper:**


- The authors develop the NODE-GAM and NODE-GA$^2$M architectures by modifying the previously-developed NODE architecture with constraints and a gating mechanism to ensure that, both within each tree and across layers, the model is only allowed to learn feature interactions of order 1 (for NODE-GAM) or 2 for (NODE-GA$^2$M).
- They show that their proposed NODE-GAM and NODE-GA$^2$M architectures perform comparably to the baselines of Explainable Boosting Machines, tree-based GAM/GA$^2$M and the traditional spline-based GAMs on medium-sized datasets while outperforming them on larger datasets. They also show that NODE-GAM can benefit from self-supervised learning by pretraining a model to reconstruct the original input from a masked input.
- They apply NODE-GAM and NODE-GA$^2$M to real-world datasets and showcase the patterns uncovered.

**Summary Of The Review:**

Overall, I think a lot of careful thought went into this paper and the design of the NODE-GAM and NODE-GA$^2$M architectures is clever enough to be inspiring in its own right. However, the missing comparisons, particularly to the "univariate networks" for modeling main effects from the "Neural Interaction Detection" paper (which is the first thing that tends to come to mind when I think of a "neural network based GAM"), leave me hesitant to recommend the paper outright. I hence currently rate it as only marginally below the acceptance threshold, but expect that can easily change on revision.

---

> ### Author Response · Authors · 2021-11-16
> **Thank you for your relevant works and detailed reviews**
>
> > The biggest weakness for me is primarily some missing baselines that I would be interested in seeing. In reading this paper, I felt there was an "elephant in the room" question lurking in my mind about why a much simpler neural-network-based architecture, such as the "univariate" networks present in the "Neural Interaction Detection" (NID) paper for modeling main effects (Fig 2 in https://arxiv.org/pdf/1705.04977.pdf; not cited) was not applied. I believe this is similar to the approach taken in the NAM paper as well (https://arxiv.org/pdf/2004.13912.pdf; cited). Regarding the NAM paper, the authors write "NAM requires training on tens to hundreds of neural networks and requires an extensive hyperparameter search...in addition, NAM's shape graphs are too smooth after bagging". As a reader, having seen the "univariate networks" used in the NID paper (which, to my knowledge, were not trained with tens to hundreds of networks), I'm left wondering "what happens if you use fewer neural networks and don't use bagging"? I'm sure other difficulties arise in this case - perhaps the model used in the NID paper is also not good at modeling abrupt changes - but it would be helpful to the reader to showcase to what extent this is an issue in practice. I get the sense that the authors are familiar with a lot of not-well-documented limitations of other approaches that led them to develop the NODE-GAM architecture, and it would be helpful if these limitations were made explicit.
>
> Thank you for bringing up the NID paper!
>
> We believe the NID paper proposed training a univariate network along with a normal MLP but did not train a univariate network alone in their paper. And given the architecture is the same as NAM (which surprisingly NAM did not mention their paper (and NIT paper), and thus we missed it as well), we think experimenting with NAM seems more appropriate just mainly because there is no GAM alone trained in NID and NAM has better released their codes and hyperparameters (NID does not provide the code about MLP-M in their github).
>
> Since NAM relies on an extensive search for hyperparameters (Table A.1, A.2), to avoid unfair comparisons we focus on 3 of their 5 datasets (MIMIC2, COMPAS, Credit) that we also use in our paper so we can directly use their reported hyperparameters. We show that without bagging the NAM performs worse than Node-GAM. Please see Appendix A in the paper.
>
> Finally, we feel that we can not argue whether the quick jumps are better than the smooth curve in all situations and whether NAM can model quick changes although their curves look very smooth, so we decide to remove this claim completely from our paper. Instead we focus on the points that (1) NAM can not model GA2M while ours can, and (2) because NAM builds a small feedforward net per feature, in high-dimensional datasets NAM may require large memory and computation. And (3) we compare with NAM without bagging and show that our model is better. Thank you for bringing this up that makes the paper stronger.
>
>
> > Still on the subject of prior work/baselines, I am interested to hear how this approach compares to the "Neural Interaction Transparency" (NIT) approach (https://proceedings.neurips.cc/paper/2018/hash/74378afe5e8b20910cf1f939e57f0480-Abstract.html) - it does not seem as though NIT is cited in this work. To quote from the abstract of NIT: "NIT is also flexible and efficient; it can learn generalized additive models with maximum -order interactions by training only models" - so it seems quite topical.
>
> Thank you for pointing this very relevant work to us. Since they do not release their code and their methods adopt some important and non-trivial modifications, we feel we can not reproduce their methods faithfully. We sent them an email asking but we're not sure they will reply since the paper has been 3 years old.
>
> One potential downside of NIT is that they have to train the network twice. They first train the network to learn which main and interaction terms survive under the soft L0 penalty, and then re-train and re-initialize the model using only those terms. And their performance is slightly lower to DNNs while ours are overall on par with it. They also do not perform purification that makes GA$^2$M graphs unique when interpreting GA$^2$M.
>
> > There are a few very noticeable typos scattered around (e.g. "Mathamatically" at the bottom of page 2, "distirbution" at the top of page 5).
>
> Sorry about it. We reread several times and we believe we correct all the typos.
>
> > In equation 7…
>
> > I was a bit confused by …
>
> You are right. We update the notations and the embarrassing “mod” operations. Thank you.
>
> > In the section on regularization, ...
>
> I do not know why I have not thought about it before. I will make sure to change it when releasing the code. Thank you!

---

> > ### Comment · Reviewer_L13g · 2021-11-18
> > **Thanks for the response - a brief remark/question/request re the NAM comparison**
> >
> > Thanks for the added comparisons. I agree the authors should not be forced to compare to previous work that has not released its code in a usable way, and I agree with the point that NAM would struggle to scale to high-dimensional datasets. I had one remark re this line in the author response:
> >
> > "We believe the NID paper proposed training a univariate network along with a normal MLP but did not train a univariate network alone in their paper. And given the architecture is the same as NAM (which surprisingly NAM did not mention their paper (and NIT paper), and thus we missed it as well), we think experimenting with NAM seems more appropriate just mainly because there is no GAM alone trained in NID and NAM has better released their codes and hyperparameters (NID does not provide the code about MLP-M in their github)."
> >
> > One thing that stood out about the NAM paper: in that paper they argue that it's important to "model jagged shape functions" and that ReLU networks have a "bias towards smoothness" that makes it difficult for them to learn these jagged shape functions; they then propose the "exp-centered (ExU) hidden units" as a solution, where they "simply learn the weights in the logarithmic space with inputs shifted by a bias", which makes the units "able to change its output significantly, with a tiny change in input". They then say "To avoid overfitting when fitting NAMs with ExUs, we employ various regularization methods including dropout, weight decay, output penalty, and feature dropout". Is the bagging of models a way for NAM to combat overfitting? I was not sure myself - I did not read the whole NAM paper in detail but I did a search for "bag" in the NAM paper and didn't find a description of how they used it. But if so, can the authors include a comparison to NAM without bagging and without the ExU units? I think that would be most comparable to what was done in the NID paper and what most people might intuitively try when they think "neural network GAM".

---

> > > ### Author Response · Authors · 2021-11-18
> > > **Thanks for the interesting idea and the quick response**
> > >
> > > Sorry I should have said the "ensemble". By looking at their code they don't actually do the bagging, but just train multiple models with different initializations.
> > >
> > > Yes we believe ensemble is part of the regularization. Qualitatively we can see in their Figure A.1 that their NAM with ExU after ensemble looks much smoother.
> > >
> > > We show the NAM with normal activations in MIMIC2 and Credit in the following table (COMPAS already uses the normal units). We find the number of "NAM-normal (Ensembled)" from the NAM's paper (Fig4 caption, and the end of page 8).
> > > We find the performances between the two units are similar in MIMIC2 but different in Credit.
> > > And in 3 out of 5 datasets (COMPAS, FICO, Housing) NAM's authors find normal hidden units perform better as their best hyperparameter (see Appendix A.1, A.2). So we guess normal units might perform equally well in most cases.
> > >
> > > The other thing we find is ensembling help most for ExU but not so much for normal units. Intuitively it makes sense - ExU is a much low-bias high-variance model that benefits more from ensemble but not so much for standard units. We can find the same pattern across MIMIC2, Credit and COMPAS.
> > >
> > > |                        | MIMIC2 |   Credit   |   COMPAS   |
> > > |------------------------|:-------------:|:----------:|:----------:|
> > > |       NAM-normal       |   82.7 (0.8)  | 97.3 (0.8) | 73.8 (1.0) |
> > > | NAM-normal (Ensembled) |      82.9     |    97.4    | 73.7 (1.0) |
> > > |         NAM-ExU        |   82.4 (1.0)  | 97.5 (0.8) |            |
> > > |   NAM-ExU (Ensembled)  |   83.0 (0.8)  | 98.0 (0.2) |            |
> > >
> > > We will update the wording bagging to ensemble in the pdf shortly. But we're not sure we should put the NAM-normal number in the paper given that's a part of NAM's hyperparameter selection. Please let us know if you think we should include those.

---

> > > > ### Comment · Reviewer_L13g · 2021-11-19
> > > > **Makes sense, score increased**
> > > >
> > > > Thanks for the response. Regarding “we're not sure we should put the NAM-normal number in the paper given that's a part of NAM's hyperparameter selection”, I think having the results available in this discussion thread is probably sufficient for people who end up with the same line of questions I had. I personally do think it’s worth including in the supplement since you have already done the work, but the reasoning for including it would have to be explained.  You could say something like “To be thorough, we explored whether removing ensembles addressed the issue of NAM being excessively smooth. We found … To be extra careful, we also considered whether removing ensembling would disproportionately impact NAM due to NAM’s inclusion of ExU activations as a hyperparameter choice (ExU activations are more prone to overfitting). Thus, for datasets where ExU was chosen as the activation, we also included a comparison to NAM with normal activations and without ensembling; such an architecture is most similar to the ‘univariate networks’ used in the NID paper. The results are shown in … Note that, for COMPAS, normal activations were already chosen in the hyperparameter search.” Up to you!

---

> > > > > ### Author Response · Authors · 2021-11-22
> > > > > **Thank you**
> > > > >
> > > > > We decide to include the result and update the paper. We sincerely thank you for making this paper stronger!

---

### Author Response · Authors · 2021-11-16
**Summary of the revision**

We thank reviewers for their very good feedback on the paper!

We summarize the major changes we made in the paper:
1. Highlight that NAM can not model GA2M in the Intro, and we run the NAM without and with the ensemble and show our NODE-GAM is better in Appendix A.
2. Mention NIT paper in the Introduction
3. Add the pseudo-code in Appendix B
4. Add a conclusion paragraph
5. We also compare NAM with normal units v.s. NAM with ExU units. We find they provide similar accuracy, and NAM with ExU will improve more from bagging.

Reviewers can see what's the difference by the revision button above.
Click on the "show revisions" -> Click on "Compare revisions" -> Click the first one "Blind Submission by Conference" and the second one "NODE-GAM: ..." -> Click "View Differences".

We respond to each reviewer individually.

---

### Decision · Program_Chairs · 2022-01-20

**Decision:**

Accept (Spotlight)

**Comment:**

The paper proposes two new generalized additive models (GAM) based on neural networks and referred to as NODE-GAM and NODE-GA2M. An empirical analysis shows that the proposed and carefully designed architectures perform comparably to several baselines on medium-sized datasets while outperforming them on larger datasets. Moreover, it is shown that the differentiability of the proposed models allows them to benefit from self-supervised learning.

Reviewers agreed on the technical significance and novelty of the proposed models and valued the clever design of the new architectures. Most concerns and open questions could be answered in the rebuttal and by changes in the revised manuscript. Based one the suggestions of one reviewer new experiments comparing the proposed models to NAM were added, which improved the paper further.  The paper should be accepted.